# Varying relationships between fire radiative power and fire size at global scale

Pierre Laurent[1], Florent Mouillot[2], Maria Vanesa Moreno[2], Chao Yue[3], Philippe Ciais[1]

[1]Laboratoire des Sciences du Climat et de l'Environnement (LSCE), CEA-CNRS-UVSQ, UMR8212, Gif-sur-Yvette, France
[2]UMR CEFE 5175, Centre National de la Recherche Scientifique (CNRS), Université de Montpellier, Université Paul-Valéry Montpellier, Ecole Pratique des Hautes Etudes (EPHE), Institut de Recherche pour le Développement, 1919 route de Mende, 34293 Montpellier CEDEX 5, France
[3]State Key Laboratory of Soil Erosion and Dryland Farming on the Loess Plateau, Northwest A&F University, Yangling, Shaanxi 712100, PR China

*Correspondence to*: Pierre Laurent (**pierre.laurent@lsce.ipsl.fr**), Florent Mouillot (**florent.mouillot@ird.fr**)

**Abstract.** Vegetation fires are an important process in the Earth system. Fire intensity locally impacts fuel consumption, damage to the vegetation, chemical composition of fire emissions but also how fires spread across landscapes. It has been observed that fire occurrence, defined as the frequency of active fires detected by the MODIS sensor, is related to intensity with a hump-shaped empirical relation meaning that occurrence reaches a maximum at intermediate fire intensity. Raw burned area products obtained from remote-sensing can not discriminate between ignition and propagation processes. To go beyond burned area and to test if fire size is driven by fire intensity at global scale as expected from empirical fire spread models, we used the newly delivered global FRY database which provides fire patch functional traits based on satellite observation, including fire patch size, and the fire radiative power measures from the MCD14ML dataset. This paper describes the varying relationships between fire size and fire radiative power across biomes at global scale. We show that in most fire regions of the world defined by the the GFED database, the linear relationship between fire radiative power and fire patch size saturates for a threshold of intermediate intensity fires. The value of this threshold differs from one region to another, and depends on vegetation type. In the most fire-prone savanna regions, once this threshold is reached, fire size decreases for the most intense fires, which mostly happen in the late fire season. According to the percolation theory, we suggest that the decreasing of fire size for more intense late season fires is a consequence of the increasing fragmentation of fuel continuity along the fire season and suggest that landscape-scale feedbacks should be developed in global fire modules.

## 1 Introduction

Fire is a major perturbation of the Earth system, which impacts the plant biomass distribution and vegetation structure, the carbon cycle, global atmospheric chemistry, air quality and climate (Bowman et al. 2009). Fire is therefore recognized as an essential climatic variable (GCOS 2011), and the potential impact of global warming on drought severity and fire season length is a key scientific question (Flannigan et al. 2009, Krawchuk et al. 2009, Aragão et al. 2018) to understand its role within the Earth system. Most Dynamic Global Vegetation Models (DGVMs) have included fire modules (see Hantson et al. 2016, Rabin et al. 2017 for a review) to improve the prediction of the impact of fire on vegetation dynamics and the carbon cycle. Substantial efforts have been devoted in the past decades to create reliable burned area (BA), active fires and fire radiative power (FRP) global datasets which allow to quantify the fire perturbation since the beginning of the 2000's (Mouillot et al. 2014) and benchmark DGVM fire modules.

A fire can be decomposed as a two-step process, the ignition and the propagation (Pyne 1996, Scott et al. 2014). Potential fire ignitions are set by lightning strikes and humans (deliberately or accidentally), and the probability that an ignition turns into a spreading fire event mainly depends on fuel type and its moisture content at the location of the ignition. The

Rothermel's equation (Rothermel 1972) has long been used to model fire propagation in landscape fire succession models (Cary et al. 2006), whose rate of spread scales with a power function of the wind velocity, landscape slope and fire intensity. However, this model, used by processed-based fire modules in most DGVM, has only been benchmarked on experimental and localized fires, discarding topographic and landscape effects. Besides, for larger natural fires, the continuity of the fuel bed also has an impact on fire propagation: a homogeneous fuel bed usually promotes fire propagation (Baker et al. 1994)

while fragmented landscape with a heterogeneity of fuel patches reduces fire spread (Turner et al. 1989). On the other hand, the velocity of fire propagation determines the amount of fuel entering the combustion zone, and therefore feeds back on the intensity of the fire event. In addition to its coupling with fire propagation, fire intensity also significantly impacts the chemical composition of the emissions (Tang et al. 2017), the amplitude and severity of vegetation damage and its post-fire regeneration ability (Bond and Keeley et al. 2005). As a result, analyses focusing on fire patch properties, such as fire patch

size and shape, rather than on simple BA have emerged in the last decade. Information on the fire patch size distribution (Archibald et al. 2010, Hantson et al. 2015, Laurent et al. 2018) can be used to map the different fire regimes at global scale (Archibald et al. 2013), and edge effects could reveal landscape scale processes leading to the observed shapes of burned patches (Greene et al. 2005, Cary et al. 2009).

Recent studies (Pausas and Ribeiro et al. 2013, Luo et al. 2017) have shown that fire occurrence, defined as the number of remotely detected active fires in unit of time per unit area, increases with fire intensity up until a threshold is reached (so-called Intermediate Fire Occurrence-Intensity (IFOI) hypothesis) above which occurrence decreases with increasing intensity. Since ignition and propagation are different processes and are not driven by the same climatic variables, it is necessary to go beyond fire occurrence and BA and to consider individual fire events. Here we document and investigate the

relationship between fire patch size derived from BA data and FRP at global scale based on remote sensing information. FRP measures the energy emitted through radiative processes released during the combustion, and can be associated with fire intensity all along the fire burning process (Wooster et al. 2005, Ichoku et al. 2008, Barrett and Kasischke 2013, Wooster et al. 2013). A positive relationship between fire patch size and the reaction intensity of the fire front is expected at least for small fire size, whose propagation rate has been benchmarked using laboratory experiments. But we do not know if this

holds up at global and regional scale for bigger fires, usually reaching longer temporal scales with varying wind directions and atmospheric circulation, and larger spatial extent. Fire patch size may not continue to increase with fire intensity above a certain size due to landscape fragmentation could act as a natural barrier against fire propagation. To uncover the fire size-intensity relationships, we assembled the information on fire patch size recovered from the FRY global database (Laurent et al. 2018) based on the MODIS MCD64A1 and the MERIS FireCCI41 burned area products, with FRP using active fire pixel

data from the MCD14ML dataset.

**2 Data and Methodology**

We used the FRY database containing the list of fire patches characterized by their morphological traits, including fire patch size, at global scale (Laurent et al. 2018). Fire patches were derived from the MERIS fire_cci v4.1 (later called FireCCI41, Chuvieco et al. 2016) and the MCD64A1 Collection 6 (Giglio et al. 2016) BA pixel products. The FireCCI41 product

provides the pixel burn dates for the period 2005-2011 and is derived from the ENVISAT-MERIS sensor, with a spatial resolution of 300x300m and a 3-day revisit frequency at the equator. The MCD64A1 product, derived from the MODIS sensors, provides pixel burn dates at global scale over the period 2000-2017 with a coarser resolution (~500x500m) but a more frequent revisit time (1 day at equator). The pixel burned dates are combined using a flood-fill algorithm (Archibald et al. 2009), which is parametrized by a cut-off value. This cut-off value corresponds to the maximum time difference between

the burn date of neighbouring pixels belonging to the same fire patch. These global datasets have been thoroughly compared

by the authors of the FRY database, locally compared using North America Forest Service fire patch database (Chuvieco et al. 2016) and validated against Landsat fire polygons in the Brazilian cerrado (Nogueira et al. 2017). The FRY database is organized in 8 datasets (2 surveys times 4 cut-off values), and provides for each individual fire patch a set of variables, called fire patch functional traits, including the geo-location of the patch centre, the fire patch size (later called FS, in hectares), and different indices on fire patch morphology. Standard Deviation Ellipses (SDE) are fitted by Laurent et al. over each fire patch larger than 5 pixels (using the "aspace" R package),and the geo-location of their centres, half-axes and orientation in longitudinal/latitudinal coordinate system are also provided for each fire patch, as well as the values of the minimum, mean and maximum pixel burn dates..

Active fire pixel data from the MCD14ML dataset (Giglio et al. 2006) consists in a list of geographic coordinates of individual active fire pixels detected by the Terra and Aqua sensors onboard the MODIS satellite for the period 2000-2017 with a resolution of 1x1km. For each pixel, the dataset provides the date and hour of burn of the active fire pixel, along with its FRP (in MW). FRP represents the energy emitted by fire through radiative processes (i.e. the total fire intensity minus the energy dissipated through convection and conduction) over its total area. It is widely used as a proxy for fire impact assessment (Barrett and Kasischke 2013, Sparks et al. 2018), biomass combustion rates (Roberts et al. 2005) or fire event (Hernandez et al. 2015) and fire spread (Johnson et al. 2017) modelling. We performed a spatio-temporal matching between active fire pixel data and all the fire patches from the FRY database in order to recover the average FRP for each fire patch. To do so, we consider that an active fire pixel belongs to a fire patch if it fulfils the two following conditions:

- The centre of the active fire pixel must be located within the SDE of the fire patch. Since the side of an active fire pixel is 1km, we also consider that an active fire pixel located at a distance of 1km or less from the area covered by the SDE belong to the fire patch.
- The detection date of the active fire pixel must lie between the minimum minus a 30 days buffer and maximum burn date of the BA pixels of the fire patch. The 30 days extension is used to account for the possible time lag between the detection of an active fire pixel and its associated burned date pixels.

Once the active fire pixels belonging to each fire patch were obtained, we compute for each patch the mean FRP value of all associated pixels. The spatio-temporal matching sometimes fails to recover any active fire pixels for some fire patches. Such fire patches (~20-25% of each sample) were discarded from the analysis. We observed that the number of fire patches without attributed active fire pixels raises as the cut-off decreases (see Supplementary Tab 1). This can be explained by the fact that, for low cut-off values, a real fire event can be split by the flood-fill algorithm in different smaller fire patches. Using a shorter value for the temporal buffer (10 days) slightly raises the failure rate of the matching, but had no significant impact on the results presented in this analysis.

In the following, we studied the relationship between FRP and FS in each region defined by the Global Fire Emission Database (GFED, Giglio et al. 2013, Supplementary 1). Since different vegetation types can occur within a GFED region (and consequently different amount of biomass or drought severity), we split all of them in three vegetation types using the GLCF MODIS Land Cover data (Channan et al. 2014) and explore the relationship between FRP and fire size for each vegetation type in each GFED regions. The vegetation types are defined by grouping together MODIS Land Cover categories: "forests" stands for all the forested land cover types (evergreen/deciduous needleleaf/broadleaf forests and mixed forests), "savannas" for savannas with woody savannas, and "grasslands/shrublands" stands for grasslands with open and closed shrublands. The spatial extent corresponding to these three vegetation types can be found in Supplementary Figure 2.

In each 1º×1º cell, we split the fire season into three periods: early, corresponding to the 4 months before the month with the highest BA, middle, corresponding to the peak BA month, and late fire season corresponding to the 4 months after the peak BA month. We did not split the fire patch distribution in different FRP categories, because of the big asymmetry of the number of fire patches between high and low intensity fires. For each period, following the same methodology as in Laurent et al. 2018, we fitted a power law against the fire patch size distribution to estimate the power-law slope parameters $\beta_{begin}$, $\beta_{middle}$ and $\beta_{end}$. These β parameters allow to investigate the asymmetry of the fire size distribution in each cell. High β values implies that the size distribution is dominated by small fires.

The results presented below have been computed for each of the 8 different fire patch datasets of the FRY database. However, we will further only focus on the results obtained from the MCD64A1-derived fire patch dataset, with a cut-off value of 14 days. The figures obtained for the FireCCI41 fire patch product with a cut-off of 14 days (which span the years 2005 to 2011) can be found in Supplementary. The same analysis was also performed with a cut-off value of 3 days for both MCD64A1 and FireCCI41: testing another extreme cut-off value allows us to estimate the impact on the results of the temporal threshold parameter used to reconstruct fire patches by Laurent et al. (2018).

## 3 Results

The median FS and median FRP are displayed on Figure 1. Large and intense fire patches are located in Australia, in the grasslands of Kazakhstan, in Namibia, in Sahel, and in Patagonia. High mean FRP values are also reached in South Australia, in the Mediterranean Basin and in the forested areas of Western USA and boreal North America. On the contrary, fires are both smaller and less intense in croplands of North America, Europe and South East Asia, and in African savannas. The fraction of BA in the cell each year is also displayed.

The relationships between the median, 25th and 75th quantiles of FS based on MCD64A1 with a cut-off value of 14 days, and FRP for different GFED regions are shown in Figure 2. The color of the dots and error bars represents the average of the minimum burn dates of the fire patches in each bin of FRP, and the background histograms the number of fire patches in each FRP bins. In all GFED regions, the number of fire patches peaks at low to intermediate FRP values (~20-30 MW). In most of GFED regions, we note that median FS and quantiles decreases once a FRP threshold is reached (Figure 2). In order to smooth the estimation of this FRP threshold (later called $FRP_{MAX}$) above which FS seems to saturate, we fitted a four-degree polynomial function to the data and determined the FRP at the maximum median FS value of the fit. The results are displayed in Table 1.

Northern Hemisphere Africa (NHAF), Equatorial Asia (EQAS) and Southeast Asia (SEAS) experience a humped relationship between FS and FRP. At low FRP values (30 to 80 MW), the median and quantiles of FS increases with FRP and reaches a maximum value at low to intermediate FRP (Table 1, Figure 2). We also identified in Figure 2 that the fire patches associated with intense fires having a FRP above the regional threshold tend to occur later in the fire season. In Central America (CEAM), Northern Hemisphere South America (NHSA), Southern Hemisphere Africa (SHAF), Southern Hemisphere South America (SHSA), and Australia (AUST), but also in Boreal Asia (BOAS), the relationship between the median and quantiles of FS vs FRP is similar. However, the maximum FS is reached at higher FRP values (from 75 to 125MW) than for NHAF, EQAS and SEAS, and the decrease following the maximum FS is more gradual. Intense fire events also appear later in the fire season for BOAS and AUST, and AUST exhibits the highest FS/FRP slope (9.0 ha.MW$^{-1}$ compared to 0.6 to 4.4 ha.MW$^{-1}$ for other regions). By contrast, in Boreal North America (BONA), Temporal North America (TENA) and Europe (EURO), and Central Asia (CEAS), mean FS constantly increases with FRP and only reaches a plateau

at very high FRP (~196 MW for BONA, ~215 MW for TENA, ~240 MW for EURO and 277 MW for CEAS). In those temperate and boreal regions, we did not observe the humped shape relation with a decrease of FS for high FRP that occurs in other GFED regions (Figure 2). Middle East (MIDE) also displays a positive correlation between median FS and FRP, but the statistics for intense fire events is too low to infer any significant relationship at high FRP values.

Figure 3 displays the same analysis as figure 2, but each GFED region is subdivided into 3 vegetation types (as defined in the Methodology section), allowing an overview of the contribution of each vegetation type by region. For BONA, TENA and EURO, mostly dominated by forest fires, we observe that the generic pattern obtained in Figure 2 is similar to the one observed for the 'forests' vegetation type, while the other vegetation types display a more humped-shape relationship. In tropical areas (NHSA, SHSA, NHAF, SHAF, AUST), the generic pattern observed in Figure 2 is similar to the one observed for the "savannas" and "grassland/shrublands" vegetation types, highlighting the uniform pattern in these two dominant vegetation types within the region, only differentiated by a higher median fire size for "savannas". 'Forests' vegetation types display a more linear relationship, closer to the one observed in temperate and boreal areas. In conclusion, the behavior of the relationship between FRP and FS obtained for each GFED region is actually representative of the main dominant vegetation types composing these regions, while the non-dominant vegetation types may experience another pattern. In all regions, savannas and grasslands ecosystems experience higher median fire sizes with a humped shape FS/FRP relationship, while forested areas experience a more linear relationship.

Figure 4 shows for 1ºx1º cells at global scale the month with the largest median FS, the month with the highest median FRP, and the phase shift between these two months. For most African cells, the month with highest median FRP is shifted between 3 to 6 months after the month with highest FS. These cells correspond to the regions where high burn area (Figure 1, Giglio et al. 2013) and a high density of fire patches are detected (Laurent et al. 2018). A narrower shift is observed in SEAS, northern AUST, and in the cells of South America with a slightly lower number of fire patches and lower BA. In Northern America (BONA and TENA), BOAS, and central and south AUST, no shift is observed, which means that the largest fires and the most intense fires happened concomitantly during the fire season. Some cells (mainly in Sahel and eastern BOAS/CEAS) displayed a negative shift, meaning that the most intense fires happened sooner than the largest fires.

The global maps of power-law slope parameters $\beta_{begin}$, $\beta_{middle}$ and $\beta_{end}$ (respectively for the beginning, middle and end of the fire season) are displayed on Figure 5. The $\beta$ parameters are only computed when more than 10 fire patches are available during the considered period, to ensure a sufficient number of patches in the fit. The differences between $\beta_{end}$ and $\beta_{begin}$ are also shown in Figure 5. The highest $\beta$ values (either $\beta_{begin}$, $\beta_{middle}$ and $\beta_{end}$) were mainly obtained in NHAF, northern SHAF, NHSA, SHSA and SEAS, as observed in previous fire size distribution analysis (Hantson et al. 2015, Laurent et al. 2018). In these regions, we found that the value of $\beta$ is higher at the end of the fire season than at the beginning, meaning that the proportion of small fires rises through the fire season, supporting our early results that late fire season don't get larger with increasing FRP. In AUST, the $\beta$ value remains constant all along the fire season, and increases in eastern BONA, TENA, and eastern BOAS, suggesting that later season fires are more dominated by larger fires. For other regions, the limited number of fire patches render difficult the interpretation of the evolution of $\beta$ through the fire season.

## 4. Discussion

Following the hypothesis from Rothermel's equation of fire spread, and considering that FRP can be used as a proxy of fire reaction intensity (Wooster et al. 2003, 2005), we used the global fire patch database FRY to test if high FRP fires propagate faster and are therefore systematically larger than low FRP fires. We found that this hypothesis is actually verified for low to

intermediate FRP in most fire regions and for the three defined vegetation types. We identified biome-specific FRP vs FS relationships, with FRP leading to maximum FS being higher in temperate/boreal forests, followed by grasslands, savannas and tropical forests.

In most fire-prone biomes, the positive relationship between FS and FRP does not hold for larger and more intense fire patches (Figure 2), generally occurring later in the fire season, as previously observed in Australia (Oliveira et al. 2015). This effect could be explained as follows: at the beginning of the fire season, when the fuel moisture content is still high, FRP is limited as energy is consumed by fuel moisture vaporization (Alexander 1982, Pyne et al. 1996) and consequently, rate of spread and fire size also get limited. As the fuel becomes dryer along the fire season (Sow et al. 2013, Sedano and Randerson 2014, N'Dri et al. 2018) fires become more intense and potentially propagate further. However, the propagation of larger fires can hit some limits due to the fragmentation of the fuel matrix, from intrinsic anthropogenic fragmentation, roads or grazing fields. The barriers limit FS as fires became larger along the fire season: these large fires will have a high propensity to reach these barriers. As a result, in fire regions with fragmented vegetation such as African savannas, Soth East Asia or at the interface between the amazon forest and croplands of South America, a maximum mean FS is reached at intermediate FRP (Figure 2). The FRP threshold differs however between these regions, possibly because their level of landscape fragmentation is different (Taubert et al. 2018).

If fire size would only be limited by the intrinsic structure of vegetation, we would not expect to see the decrease of the proportion of large fires toward the end of the fire season in fire-prone ecosystem (Figure 5). If the number of individual fire events is already high at the beginning of the fire season, the landscape becomes even more and more fragmented by BA scars (Oliveira et al. 2015) and fuel load decrease (N'Dri et al. 2018), meaning that the limitation of fire size due to landscape fragmentation will be higher for fires ignited later in the fire season (Teske et al. 2012). As a consequence, this mechanism may explain why the correlation between FRP and FS becomes negative in Figure 2 during the late fire season in NHAF, NHSA, CEAM, EQAS and SEAS, and why $\beta_{end}$ is higher than $\beta_{begin}$. This limitation of fire size for intense fires in those regions, possibly due to the feedback between fire and fuel connectivity at landscape level, is in line with the results obtained from Mondal and Sukumar (2016) relating the effects of recent past fires on fire hazard in dry tropical forests, and otherwise theoretically approached from the percolation model applied to wildfires by Archibald et al. (2012). This model shows that the amount of BA is maximized when both the fire spread probability and the fuel matrix connectivity are high. BA dramatically drops if fire spread probability is too low (such as in the beginning of the fire season) or if the fuel array connectivity becomes too small (such as in the end of the fire season). Particularly, the percolation model shows that BA can drop dramatically once 50-60% of the available fuel has burned, which is close to the maximum percentage of BA detected by both MCD64A1 and FireCCI41 products (Giglio et al. 2013, Chuvieco et al. 2016). The IFOI hypothesis, proposed by Luo et al. (2017) to explain why fire occurrence is limited by fire intensity, can be interpreted as a direct consequence of percolation theory applied to fire-prone ecosystems.

For regions where fire events are less frequent, such as in BONA, TENA and EURO (Figure 2), there is no significant limitation of fire spread and fire size, suggesting that the fragmentation of landscape either from land use or from early season burn scars does not limit fire spread (Owen et al. 2012). Fire size remains positively correlated with fire intensity all along the fire season. Moreover, the 75th quantiles for BONA and TENA is higher than for tropical regions (except AUST), most probably because tree species in BONA and TENA (e.g. spruce) are more flammable , because crown fires are more frequent, and because these ecosystems experience an actual drought period compared to the tropics where rainfalls occur more frequently. They can therefore propagate further than herbaceous fires hardly turning into crown fires in savannas and woodlands in semi arid tropical regions. In BOAS the relationship between FS and FRP is different from the one observed in

BONA and TENA. This could be a result from the less flammable vegetation and the highest number of ground fires in BOAS (Kasischke and Bruhwiler 2003). Moreover, BA detection of surface fires (and consequently, fire patch characterization) is known to be difficult in boreal Asia, and numerous discrepancies have been observed between the BA products obtained from different moderation resolution sensors (Chuvieco et al. 2016).

The median FS is globally lower for the datasets generated from FRY with smaller cut-off value (see Supplementary 1 and 2), because big fire patches tend to be split in smaller patches for lower cut-off values, reducing the average fire patch size. The median FS is also lower for the FireCCI41 derived datasets, due to its ability to detect smaller patches from its better spatial resolution. Changing the survey or the cut-off value does not impact the global distribution of large and small fire patches. Reducing the cut-off to 3 days does not change the observed relationship between FS and FRP. The results obtained from the dataset derived from FireCCI41 follows the same trend, but for some GFED regions (TENA, EURO, NHSA, AUST), the seasonality is shifted one month later than for MCD64A1. Reducing the cut-off values lowers the temporal shift observed on Figure 4 at global scale (Supplementary 3 and 4), but the global distribution of the shift is conserved. Similarly, FireCCI41 yields smaller shifts than for MCD64A1, but with the same spatial distribution.

In the previous section, we hypothesised that FRP can be used as a proxy of fire reaction intensity but the limitations of such an approach should be mentioned. First, the energy released by a wildfire can be decomposed in three parts: convection, conduction, and radiation. FRP only represents the radiative part of the energy released by a fire. Moreover, the fire reaction intensity used in Rothermel's equation does not share the same spatial extent as FRP: fire reaction intensity pertains to the flaming front of the fire, while FRP integrates all the radiative energy emitted over a 1 $km^2$ window. This means that radiation emitted from smouldering can also contribute to FRP, not only the flaming front. The impact should differ for different wetness conditions and vegetation types: smouldering fires are more frequent in forested areas, whereas in grasslands most of the detected radiative power will be released by the active fire front. Another issue appears from the integration of radiative energy over the 1 $km^2$ window: very often active burning fire lines do not cover the whole 1-$km^2$ area so that measured FRP is a mixed signal from both active-burning and unburned areas. However, we can expect this effect to be mitigated by the fact that our analysis does not account for very small fires, since the FRY database does not provide fire patches smaller than 107 ha for MCD64A1. Finally, a recent study (Roberts et al. 2018) used 3D radiative transfer simulations to show that the canopy structure intercepts part of the FRP emitted by surface fires. This means that the FRP measured from remote sensing for forested areas and savannas could underestimate the actual FRP. We can also expect this underestimation to vary with tree species that are associated with different fire regimes. For example, it is probable that the amount of radiation energy intercepted by the canopy differs strongly between crown fires from highly flammable black spruce and jack pine forests from BONA (Rogers et al. 2015) and surface fires from larch-dominated forests in BOAS. These facts advocate the importance to differentiate the relationships between fire size and FRP in different vegetation types with different fire regime and fire adaptations, due to varying degrees of reliability of using FRP as a proxy of fire reaction intensity.

Thresholds of FRP detection vary between 9 and 11 MW (Roberts and Wooster 2008, Schroeder et al. 2010) for the MODIS FRP products, below which reliable detection becomes impossible. In turn, analysis based on comparison with finer-resolution remote sensing products actually concluded that MODIS might underestimate by 20% the number of captured fire pixels, particularly for small fires (Wooster et al. 2012, Peterson et al. 2013). This 9-11 MW threshold falls in the first bin of the FRP histograms in Figure 2, and could therefore explain the peak of the number of fire patches at intermediate FRP (~20-30 MW). The amount of radiwive energy reaching the MODIS instruments is much smaller at larger scan angles than at Nadir. This means that the MODIS instruments will be less sensitive to low values of FRP at high latitude (Giglio et al. 2003,

Schröder et al. 2005). This could explain the difference of the distribution of FRP associated with fire patches in BONA: the stronger asymmetry of the distribution in this region (i.e. the larger tail toward high FRP values) could arise from missing active fire data from less intense fires in this region. The temporal sampling of FRP also differs with the latitudinal coordinate since the number of satellite overpasses is larger at high latitude than at the equator (from 2 observations per day until 15 at the poles, Giglio et al. 2006). This should raise the chance to recover FRP information for fire patches at high latitude, assuming that radiative intensity is high enough to exceed the higher detection threshold at larger scan angles. Also, in some regions (such as NHAF and SHAF), fires exhibit a strong diurnal cycle (Giglio et al. 2006). The detection rate of active fires will therefore be higher if the peak of diurnal intensity is synchronized with satellite overpass. However, we can expect the sampling error rate and the variation of FRP sensitivity with latitude to be more homogeneous within each GFED regions that at global scale.

Fire season length has changed over the last 50 years and is now longer in 25% regions of the world (Jolly et al. 2015). An increase of drought intensity in fire prone environment could yield to more intense fire events, yielding larger BA patches for each fire event. However, if the progressive fragmentation of landscape through the fire season limits fire size, then it can be expected that a longer fire season would only have a limited impact on the increase of BA in these regions. In the same way but on a longer time scale in less fire prone regions, previous large fires have been shown to limit FS in the recent timeframe in western US (Haine et al 2013), and previous landscape biomass composition, as a result of fire history, is a major factor affecting fire severity in boreal forests (Whitman et al. 2018). On the contrary, in regions where the quasi-linear relationship between fire size and FRP is valid even for high FRP, a longer fire season could dramatically increase burn area, particularly in North American forests (Gillett et al. 2004, Turetsky et al. 2011). This hypothesis does not account for the impact of increased severity of fire damage to the vegetation in these ecosystems, and its feedback on fire propagation and occurrence. Our results are consistent with those of Andela et al. 2017, who showed that, contrary to what would be expected from the rise of the fire danger index, BA tends to decline at global scale (25% loss between 1998 and 2015). This decline is especially strong in savannas and grasslands, because of agricultural expansion, which results in a reduction of burnable area and a more fragmented landscape (Kamusoko and Aniya 2007, Oliveira et al. 2017, Sulieman et al. 2018). Landscape fragmentation is also a tool used for fire management. Indigenous burning practices in West Africa promote early burning and therefore landscape fragmentation in order to limit large and intense fire events which could occur at the end of the fire season (Laris 2002, Laris and Wardell 2006, Le Page et al. 2015, Archibald 2016). Similarly, US forest services used artificial fuel-breaks to fragment the landscape and limit fire size (Green 1977, Agee et al. 2000), as well as fire intensity (Ager et al. 2017).

Some DGVM fire modules explicitly simulate BA as the product of individual successful fire ignitions with mean fire size (Thonicke et al. 2010, Yue et al. 2014). In these models, fire size usually depends on wind speed, fuel bulk density and fuel load. Because of the reduction of the available fuel load due to burning by preceding fires, we can expect than BA saturates toward the end of the drought season in DGVMs, but this mechanism does not account for landscape fragmentation (due either to land use fragmentation or progressive fragmentation by fires). The LPJ-LMFire v1.0 (Pfeiffer et al. 2013), a modified version of the Spitfire module for pre-industrial global biomass burning, accounted for passive fire suppression due to landscape fragmentation. Further refining of process-based fire modules would require extensive comparison with fire patch data rather than raw BA.

## 5. Conclusion

We characterized for the first time the actual relationship between fire size and fire intensity using a combination of fire patch size and active fire datasets at global scale. We found that in most fire-prone ecosystems, fire size increases with fire intensity only at low fire intensity, reaches a threshold at intermediate intensity, and then starts to decrease. On the contrary, in temperate and boreal forests, FS and FRP are proportional even for high fire intensity. This behavior is observed with significant differences between land cover types (shrublands/grasslands, savannas and forests) for both MCD64A1 and FireCCI41 products, and for all cut-off values used for fire patch reconstruction. We suggested that the FRP threshold value is influenced by the fragmentation of the landscape, and the feedback between fuel connectivity and burn area during the fire season. This fragmentation hypothesis is consistent with the percolation theory applied to fire spread. The fragmentation hypothesis should be further tested with higher resolution BA datasets, combined with fine temporal resolution land cover datasets characterizing the landscape fragmentation, associated with temporally varying fuel moisture data, and further considered in the development of fire-DGVM models. Additional information as fire shape complexity and elongation from the FRY database should bring substantial information to assert our conclusions.

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

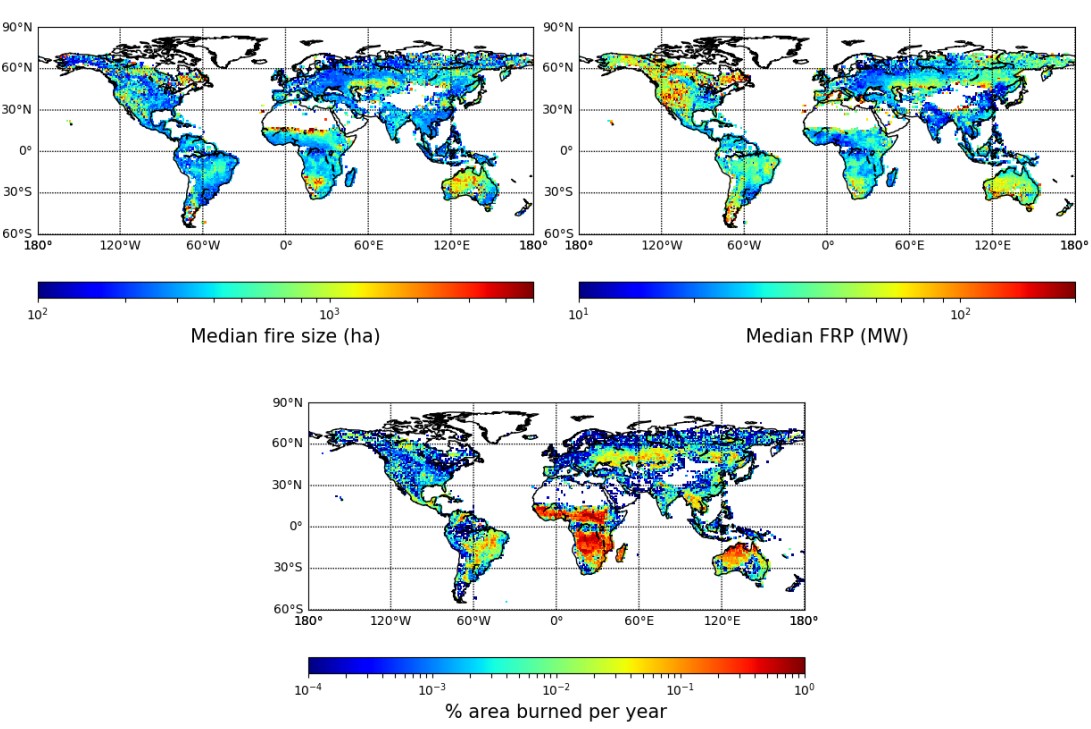

**Figure 1: Median fire size (in ha), imedian fire radiative power  from FRY database (derived from MCD64A1 with a cut-off of 14 days), and percentage of burned area each year (from GFED).**

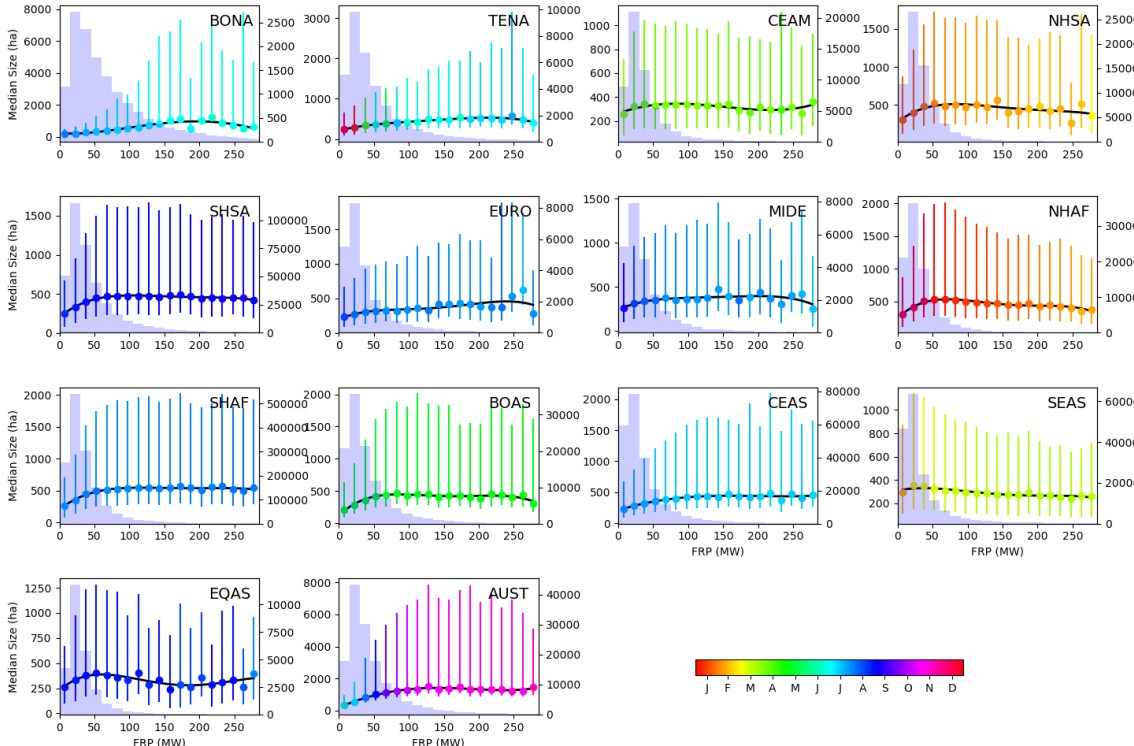

**Figure 2: Median fire size vs ifire radiative power (FRP in MW) for different GFED regions. The error bars represent the 25th and 75th quantiles of the FS distribution. The color of the dots and error bars represent the mean burn date of fire patches in each FRP bin. The black line shows the interpolated 4 degrees polynomial used to smooth the value of FRP associated with maximum median fire size. The background histograms represent the number of fire patches in each FRP bins.**

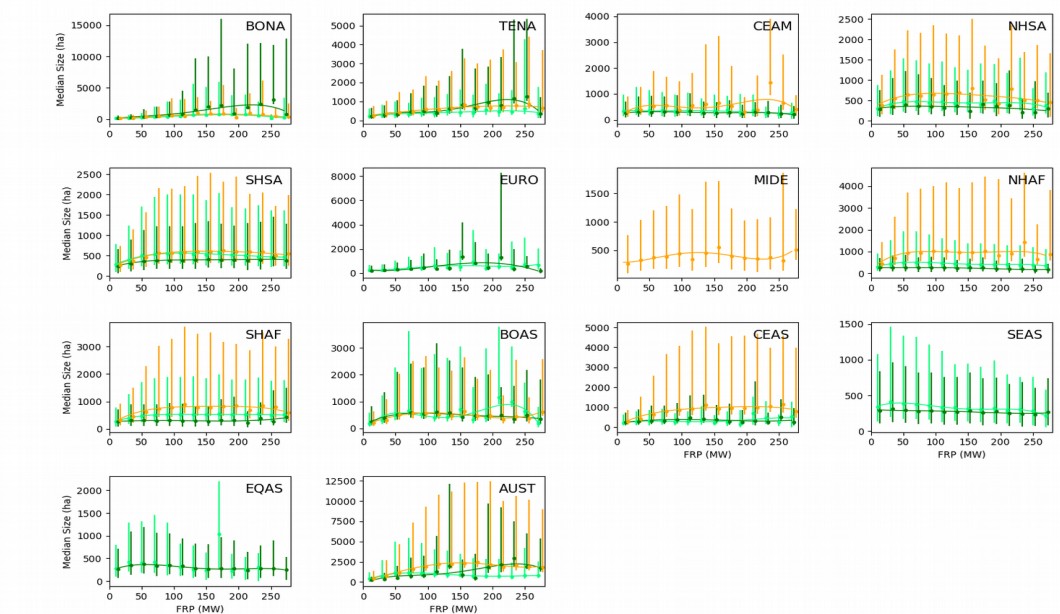

**Figure 3: Median fire size vs fire radiative power (FRP) for different GFED regions for savannas (light green), forests (dark green) and grassland/shrubland (orange). These vegetation classes are obtained by grouping similar land cover type from MODIS Land Cover data, and their spatial extent can be found in Supplementary. The error bars represent the 25th and 75th quantiles of the FS distribution. The color lines show the interpolated 4 degrees polynomial used to smooth the value of FRP associated with maximum median fire size for each land cover type.**

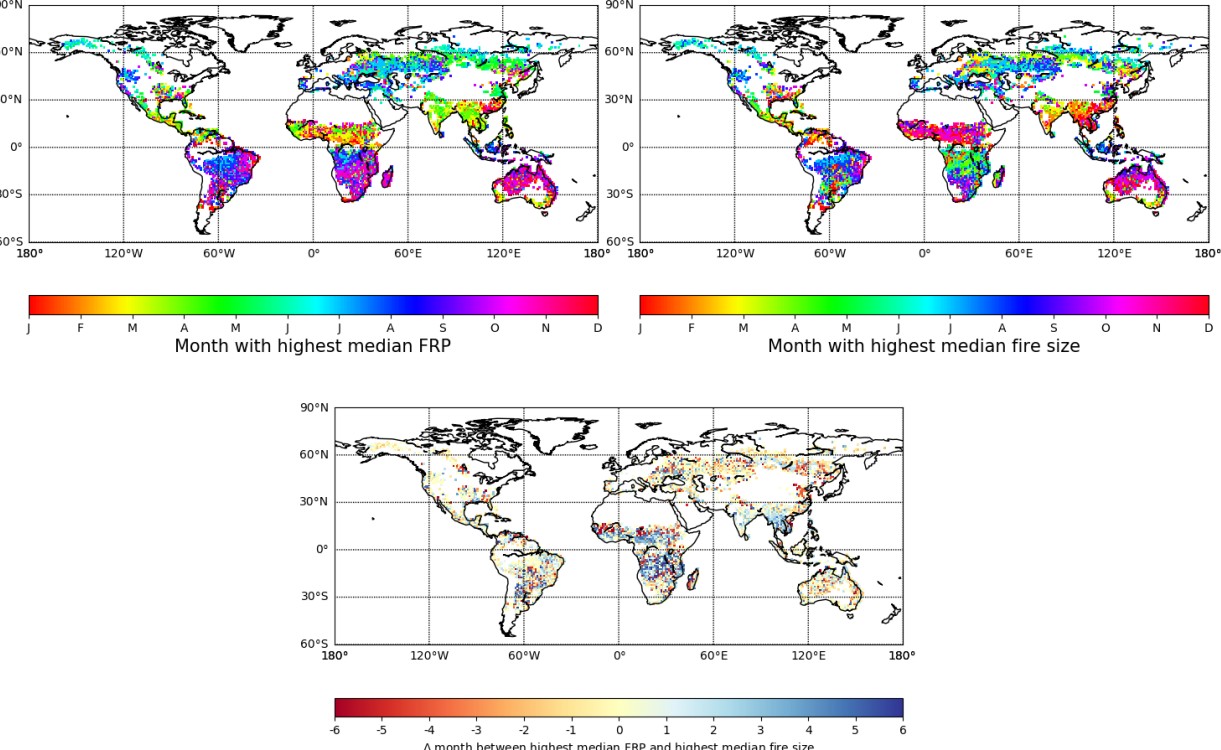

Figure 4: Month with highest median fire radiative power (FRP,top left), highest median FS (top right), and the difference between the two (bottom). In blue cells, the month with the largest fires events happen before the month with the most intense fires. In red cells, the month with the largest fires events happen before the month with the most intense fires. In yellow cells, the months with the largest fires and with the most intense fires are the same.

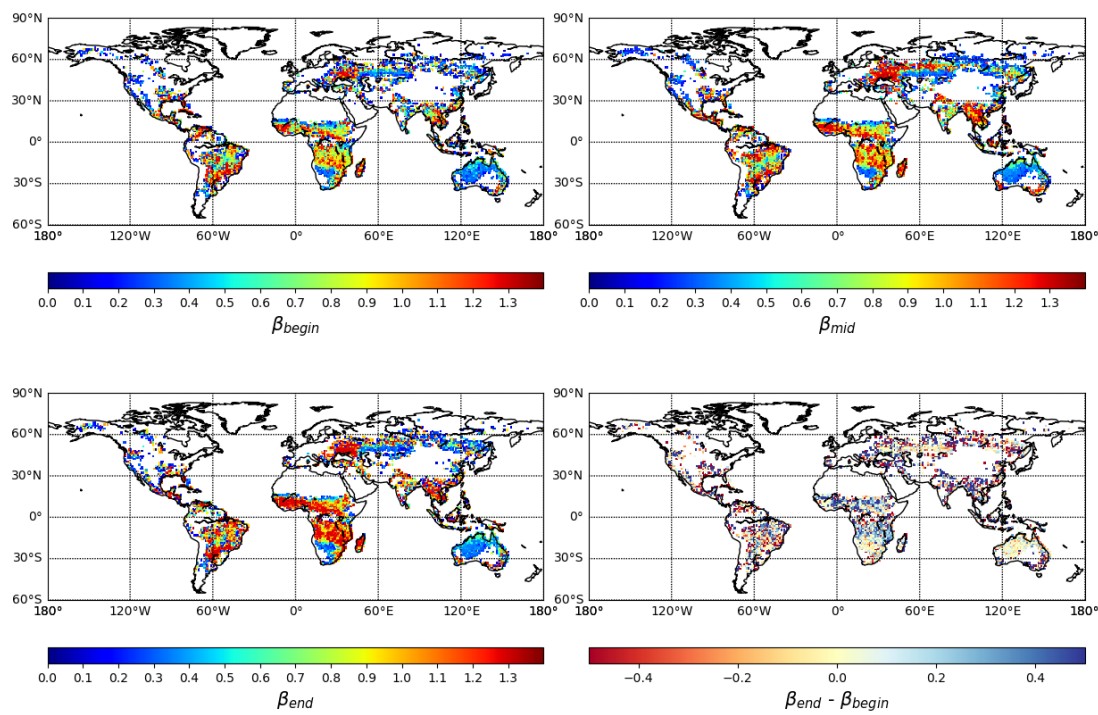

**Figure 5: Value of the log-log scale slope of the fire size distribution at the beginning of the fire season, beta (4 months before the month with the highest amount of BA), in the middle of the fire season (corresponding to the month with the highest BA) and at the end of the fire season (4 months after the month with highest BA).**

| Vegetation type | GFED Region | FRP with largest associated fire patch sizes (MW) | Slope of the FRP vs median FS relationship before max FS (ha.MW$^{-1}$) | Slope of the FRP vs median FS relationship after max FS (ha.MW$^{-1}$) |
|---|---|---|---|---|
| Savannas | BONA | 175 | 3.907 | -7.791 |
| | TENA | 221 | 1.179 | -1.974 |
| | CEAM | 10 | 2.342 | -0.499 |
| | NHSA | 74 | 2.813 | -0.400 |
| | SHSA | 110 | 2.692 | -0.661 |
| | EURO | 270 | 1.548 | NA |
| | NHAF | 67 | 5.256 | -0.662 |
| | SHAF | 110 | 2.300 | -0.172 |
| | BOAS | 224 | 2.149 | -19.993 |
| | CEAS | 260 | 0.577 | 1.011 |
| | SEAS | 45 | 3.865 | -0.575 |
| | EQAS | 185 | 1.716 | 2.038 |
| | AUST | 75 | 13.665 | -1.684 |
| Forests | BONA | 220 | 9.204 | -39.657 |
| | TENA | 222 | 3.404 | -19.811 |
| | CEAM | 57 | 1.071 | -0.382 |
| | NHSA | 76 | 1.288 | -0.457 |
| | SHSA | 242 | 0.494 | -3.859 |
| | EURO | 185 | 4.979 | -6.128 |
| | NHAF | 68 | 0.609 | -0.508 |
| | SHAF | 270 | 0.076 | NA |
| | BOAS | 88 | 5.734 | -1.075 |
| | CEAS | 90 | 1.421 | -0.696 |
| | SEAS | 10 | 3.865 | -0.224 |
| | EQAS | 55 | 2.904 | -0.395 |
| | AUST | 237 | 9.533 | -8.085 |
| Grasslands/shrublands | BONA | 170 | 5.239 | -1.579 |
| | TENA | 219 | 2.342 | -2.809 |
| | CEAM | 230 | 2.003 | -11.986 |
| | NHSA | 100 | 3.014 | -1.451 |
| | SHSA | 148 | 2.700 | -0.726 |
| | MIDE | 270 | 0.136 | NA |
| | NHAF | 220 | 1.329 | -13.382 |
| | SHAF | 170 | 2.939 | -2.049 |
| | BOAS | 105 | 5.081 | -0.402 |

|  |  |  |  |  |
|---|---|---|---|---|
|  | CEAS | 208 | 3.725 | -2.341 |
|  | AUST | 149 | 16.639 | -4.785 |
| All | BONA | 196 | 4.420 | -7.817 |
|  | TENA | 215 | 1.359 | -1.513 |
|  | CEAM | 84 | 0.775 | -0.154 |
|  | NHSA | 83 | 2.318 | -0.637 |
|  | SHSA | 105 | 2.384 | -0.237 |
|  | EURO | 239 | 0.628 | -8.143 |
|  | MIDE | 198 | 0.553 | -1.254 |
|  | NHAF | 71 | 3.939 | -0.683 |
|  | SHAF | 116 | 2.474 | -0.115 |
|  | BOAS | 86 | 3.409 | -0.346 |
|  | CEAS | 277 | 0.613 | NA |
|  | SEAS | 37 | 3.906 | -0.327 |
|  | EQAS | 60 | 3.112 | -0.187 |
|  | AUST | 142 | 9.169 | -0.523 |

**Table 1 : Value of the FRP threshold at maximum median FS, and the slope of FS vs FRP before the threshold value for different GFED regions.**

