# Peer review of "Varying relationships between fire radiative power and fire size at global scale"

_Biogeosciences, 2018_

## Short Comment (SC1) · 7 Sep 2018

This is interesting work but I detected a few issues that if addressed would make the manuscript more accurate.

L35. Why is Van Wagner cited in relation to Rothermel's model, with which he had no relation whatsoever? Van Wagner was Canadian, and so involved with the Canadian fire behaviour prediction system, not the U.S.

L36-37. "whose rate of spread scales with a power function of the wind velocity, landscape slope and fire intensity." The authors are referring to reaction intensity, not fire intensity (aka fireline intensity, which is the product of rate of spread, fuel consumption and heat of combustion and can be correlated to a certain extent with FRP).

[Figure]

L41-42. "On the other hand, the velocity of fire propagation determines the amount of fuel entering the combustion zone, and therefore feeds back on the intensity of the fire event." Not sure what this means. Rate of spread is an intrinsic component of fire intensity but not because it affects fuel consumption.

L42-43. "fire intensity also significantly impacts the fuel combustion completeness". It's the other way around, fuel consumption is an element in the calculation of fire intensity.

L57. This is general, i.e. not specific of Rothermel's model. For given fuel conditions/ fuel types faster fires are more intense, and faster fires will become large.

L95. Has fire intensity been defined?

L170. The hypothesis does not stem from Rothermel's model, it just happens that fire intensity by definition (Byram 1959) is the product of rate of spread, fuel consumption and heat of combustion, as mentioned before.

L221. "They can therefore propagate further than ground fire and fire resistant species found in savannas and woodlands". This sentence is confusing. Fire in savanna is driven by grass, not by trees (which are resistant only in the sense that they are fire adapted).

I think the interpretation of the findings, by being concentrated on the effect of fuel connectivity, is restrictive. The authors could improve the discussion by considering that the most powerful driver of fire spread/size is wind speed (see the switches of Bradstock 2010). Thus, fuels can be totally available to burn due to drought, and produce intense fires that are not that large because they do not coincide with strong winds and low relative atmospheric humidities. Thus, the annual cycle of fire extent and intensity is also a matter of timing of coincidence between drought and atmospheric conditions.

References:

Bradstock RA. 2010. A biogeographic model of fire regimes in Australia: current and

future implications. Global Ecology and Biogeography 19: 145–158.

Byram, G.M. 1959. Combustion of forest fuels. In 'Forest fire: control and use'.(Ed. KP Davis) pp. 61–89.

---

## Referee Comment (RC1) · Anonymous Referee #1 · 13 Sep 2018

Laurant et al. provide a novel data analysis based on a recently published database of fire parameters derived from remote sensing data. So far mostly burned area data were investigated to understand the global "occurrence" of fire. However burned area is not always the most informative and important parameter, fire size and fire intensity might better capture the impact of the specific fire event on ecosystem and society. The work presented here is therefore an important step. I have a couple of comments, most of them minor, that will hopefully help to improve the manuscript, but I also have two major concerns:

The first is that the study is based on the GFED regions but then wants to interpret differences between the regions to be driven by biomass availability and drought. If

you want to understand whether differences are driven by biomass and drought then a much more straight forward way to analyse the data would be to group them according to biomass and drought, not according to the GFED regions that average over all Northern Hemisphere Africa, which contains the whole gradient from desert with low biomass and strong drought to tropical rainforest with no drought and high biomass. I am very confident that the results could be much clearer and support your conclusions much better if the study design was rearranged to directly look at the effects of drought and biomass on these relationships, by grouping the data according to these two parameters.

The second is that the manuscript needs a discussion of reliability of the data, especially for the fire intensity. There are a number of limitations on the observability of fire radiative power. The point that there are still these clear spatial and temporal patterns in my opinion indicate that there is useful information in the dataset, however the problems associated with the dataset should be mentioned and discussed. For instance the energy observed is the energy released in one pixel, this energy might come from a very intense fire covering a small part or a low intensity fire covering large part of the grid cell. The observation of fire intensity strongly depends on the scan angle. Moreover fire intensity has a diurnal cycle and peak fire intensity might differ between the biomes. The satellite overpass might happen at the peak time in some grid cells but not in others. Vegetation structure influences what the satellite can observe, intensity of sub-canopy fires will certainly be underestimated. I am not an expert in remote sensing, but I think that such issues need to be mentioned to provide a balanced discussion of the results.

Specific comments:

l.17: thresholds differ between regions: what defines the regions? climate, humans, vegetation types?

l. 20: seasonal effects, could there be an influence of anthropogenic fire use too?

Percolation theory explains why fires are most intense of why fires are smaller in the late season? I guess the latter.

l. 25-27: not sure I agree 100% with the reasoning: fire models have been included before in DGVMs, for instance Arora and Boer (2005). I think the reason was more the strong impact on vegetation and overestimation of tree cover in savannas in many DGVMs.

l. 28: prediction of vegetation dynamics and the carbon cycle.

l 47: also the impact of fire varies with the size, the fire size characteristics therefore could be more informative than only burned area.

l. 52: maybe drivers of propagation and ignition are not driven by the same climate variables, but the fraction of ignitions turning into fires is determined by similar drivers, burned area and fire counts therefore have quite similar spatial patterns.

l.57: is it fire intensity of fire line intensity? and based on the equations this is not expected for large fire size? or do you mean the Rothermel equations were only tested for small scale (laboratory to stand scale) and it is unclear whether the euqations hold true for larger scales (not for larger fires).

l. 95: explain the difference between fire intensity and fire radiative power.

l. 96: are there any spatial or temporal patterns in the the discarded fire patches? This might indicate biases in the FRP detection.

l.110: text says median, figure caption says mean. The figure could also include a burned area map to show that the patterns are different, between the characteristics.

l.112: patterns of size and FRP look not so similar to me.

l. 117: use either mean or average.

l.119: could this peak simply be because lower intensity is simply not detected by the
satellite. What is the explanation for this peak at intermediate fire intensities?

l.121: change "number individual.." to " number of individual..." I assume fire counts is related more closely to burned area as two counts could be individual fires or the same fire, so some differences are also expected.

l.124: It would be useful to consistently use FI or FRP, now it is FI in the text and FRP in the figure.

l. 125-28: I don't see that the fire size is clearly decreasing. it is a bit tempting also to interpret the error bars as error bars. Maybe having three lines for 25th percentile, median 75th percentile could avoid that misunderstanding. Probably showing the 4th order polynomial with uncertainty bands could give a better impression whether decreases and maximum are robust. My confidence based on the plots shown is rather low, and now these threshold become quite important for the following discussion. Showing some kind of robustness and uncertainty on this threshold would therefore be important.

l. 174: hihger FI threshold for forests: can this be explained by Rothermel?

l. 176-7: Rothermel also uses different parameters for different vegetation types and fuel moisture and is able to reproduce the varying constraint hypothesis.

l.185: GFED regions are not biomes

l.195: so you expect a lower threshold for higher fragmentation? is that what you find in your analysis?

l.215: I would expect a very high fragmentation in EURO and TENA (lots of big streets) and strongly managed, which is why fire models usually overestimate burned area there. Is this only meant for interpreting the seasonality? so no fragmentation due to burning?

l.220: not all the tropics has rainfall all year long.

l. 221: are you suggesting that the savanna species suppress fire? burned are is much higher in savannas than in TENA and BOAS. I don't understand the logic here.

l. 223: vegetation is less flammable where?

l. 237: FDI increased everywhere?

l.251: agricultural expansion leads to a reduction of burnable area. why? croplands are burned, pastures are burned. Also the more fragmented landscape, is there a study showing that the landscape is more fragmented. I is an assumption in models and to explain the decline in burned area. Give a reference where this fragmentation is observed, or identify it as a common assumption. Could this be an effect of having smaller fires in croplands and therefore the detectable burned area is declining, not the burned area itself?

l. 259: BA saturates toward the end of the drought season: is this really reproduced by models? Any reference?

l. 267: fire-prone ecosystems: actually you didn't group the analysis by ecosystmes, for the tropical regions you group everything together, tropical rainforest and savannas are not separated. I think it would be smarter to group the data for this analysis based on vegetation and climatic parameters not by geographical regions. grouping high and low tree density together could confound the results.

l. 271: FI threshold is driven by biomass and drought severity: Most of the regions have a strong variation of biomass and drought severity. It therefore would be better to use drought severity and biomass to group the data.

---

## Referee Comment (RC2) · Anonymous Referee #2 · 14 Sep 2018

This manuscript is interesting and fit well with the focuses of BG. It can be published after a careful revision. I am not a fire ecologist. Consequently, I met a lot of difficulties in understanding concepts, variables names and their definitions you used in the manuscript. The guiding principle of your analysis is the Rothermel (1972)'s fire spread model ("Rothermel's equation" line 35, "Following the hypothesis from Rothermel's equation of fire spread" line 170). It is a very detailed local scale model. It is one of the most used models to simulate the forward rate of spread at the front of a surface fire, and is the primary fire spread model applied in many fire prediction systems. In the Rothermel's model, rate of spread is simulated as a function of topography, microclimate conditions and a fire behavior fuel model or fuel model that consists of numerous parameters for a given fuel complex. Standard fuel models have often been shown

inappropriate for representing local conditions. In this manuscript, you referred to the Rothermel (1972)'s equation. In the original USDA paper, the number of equations was c.a. 90. It will be fine that in your up-scaling procedure, from local to global, you explain how you summarized the Rothermel (1972)'s fire spread model for finally analyze the relationship between fire patch area and fire intensity. A short explanation will be useful and will clarify the discussion in which you mixed: fuel biomass availability, biomass gradient, moisture content of the fuel, fragmentation, wind speed, fuel bulk density, fuel load, etc. My second main concern is your cutting of continents by using the one proposed by the GFED. The 14 regions are very arbitrary. As an example EURO includes the surrounding of the northern part of the Mediterranean Sea where the fire regime surely doesn't follow the same pattern than in more Northern regions. Likely using a more "ecologically-based" or "climatically-related" cutting will yield contrasted results?

Line 23 plant biomass distribution. Line 25 rather ecological driver than climatic variable. Line 29 reliable burned area, active fires and fire intensity global dataset. Line 45 fire patches vs raw burn area. Please could you explain? Line 54 please define BA here (burned area). Line 62 please detail MCD14ML. Best to give the complete name of the remotely sensed products you used and their DOI if available. Line 76 fire patch size why not fire patch area? Line 74 "validated against Landsat fire polygons". Line 77 Standard Deviation Ellipse (SDE) Please could you explain how this parameter calculated? It does not seem further used in the manuscript except lines 87 and 89. One SDE covers approximately 68 percents of the fire patch. You applied a cutoff at SDE + 1 km, why not 2 SDE? Line 90 30-day buffer seems very long. During this delay surface reflectance may drastically change with resprouter shrubs or some bunchgrasses. Line 95 you wrote "In this analysis, we used FRP as a proxy of fire intensity, later called FI". Further we still found FRP in the text and in the graphs. Line 112 "Brazilian tropical savannas". On fig 1b, most red dots are located across Argentina and not across Brazilian tropical savannas! Line 125 please define the meaning of GFED. Please use the full names of the regions in Table 1. Line 126 fitted rather than interpolated. Line 130 humped relationships in CEAM, EQUAS, SEAS. This type of

"humped" relationships seems to occur elsewhere? You presented these three areas as equatorial biomes. This means closed to equator or with a particular climate pattern? (See my previous comment on your geographical cutting). Line 139 MW-1 Line 206 percolation or cellular automata? Figure 2 FI in the figure legend and FRP in the x-axis. Y-axis scales drastically change depending of geographic area and so complicate the reading. Figure 3 are you sure that this figure is necessary (see Table 1 content).

---

## Referee Comment (RC3) · Anonymous Referee #3 · 14 Sep 2018

Laurent et al. combine a new global dataset of fire patch sizes with observations of fire radiative power (FRP) to estimate how "fire intensity" changes as a function of fire size. The manuscript is well written. However, I do have some serious concerns about the interpretation of the data that would need to be addressed before I would recommend publication in Biogeosciences.

Major concern

The manuscript lacks a proper discussion (and references) of potential issues that may arise when estimating fuel consumption and subsequently fire intensity from FRP. FRP observations from MODIS represent infrequent snapshots of energy release across the pixel area (at best ~1km2 at nadir). This results in a number of difficulties when linking FRP to fire temperatures or -intensity- of which several will likely be a function of envi-

ronmental gradients. First, FRP is an estimate of energy release across an entire pixel, ~1 km2 at nadir for MODIS. It is very uncertain what fraction of the grid cell is actually burning (and this is likely a function of fuel loads and other aspect of fire behavior). Yet, this is a requirement to estimate fire intensity because if 1% of the pixel produces 10 MW of energy, or 50% of the pixel produces the same amount makes a difference of 50 times the "intensity". Second, several studies suggest that vegetation structure (in particular tree cover) also have a significant effect on the relationship between fuel consumption and observed FRP (e.g. Roberts et al., 2018 RSE). Third, the sensitivity of the MODIS instruments to detect active fires (i.e. minimum FRP that can be observed) is a direct function of the scan angle and is up to a factor of 5 lower at large scan angles compared to nadir. This may be important when looking at distributions (e.g. median), because you are likely to strongly underestimate the occurrence of low FRP values. Fourth, the fire diurnal cycle (a function of fuel conditions, vegetation type, and climate) also produces a sampling error, since there are only few daily overpasses and in some ecosystems fire activity may peak already early in the morning while in others this maybe later in the afternoon.

It would be important to properly discuss what "MODIS FRP" actually represents. I also disagree with the statement "This is in agreement with .. , since these quantities are two proxies of the number of ignitions." (lines 120-122). I do not see how the number of active fire detections is related to ignitions? A single fire may produce up to hundreds of active fire (FRP) detections if it becomes large enough and burns for a long period of time. Several studies have linked active fire detections (with or without FRP) to total amounts of fuel consumption (or biomass burned), which would be a function of area burned, fuel loads and other conditions. Moreover, looking at the distribution of FRP detections may become problematic here. In high fuel load temperate and boreal forested systems a large share of the active fire detections may come from smouldering rather than the active fire front (and ratios may change over the fire's lifetime), while for grasslands it may be mostly actively flaming fire fronts that are observed. In this light it would be important to much better define "fire intensity" (i.e. what do the authors want
to measure exactly?), and discuss how using FRP as a proxy for this quantity may be further influenced by the above mentioned limitations.

Minor suggestions

Line 87. That is ok, but what do you do if you have two adjacent fire patches? Are you double counting the active fire detections?

Line 95. ".., we compute for each patch the mean FRP value of all .. ". This isn't entirely clear to me, do you first estimate the mean of each patch and then look at the median across patches? Again, it would be important to understand what the distributions look like (e.g. across land cover types) to understand the potential implications of such decisions.

Line 155 "In each 1x1 cells", typo.

Lines 155 – 160, please move this to the methods section, accompanied by a short explanation on how that helps to answer your research questions.

Line 170 "Following the hypothesis from Rothermel's equation", maybe be a bit more specific here and add references. For clarity you could also repeat your own objectives here, e.g. "We aim to investigate if fire size and intensity are driven by a same set of environmental and climate conditions.." Also, I am somewhat surprised that in addition to speed, the authors don't mention fire duration as a potential driver of larger fire sizes.

Line 174 "Tropical areas" is not a vegetation type, delete?

Line 178 "experience limited fire energy" what does this mean? Do you mean to say something like "In equatorial areas with high annual rainfall, biomass burning is characterized by low spread rates are combustion completeness (cite), resulting in a more gradual release of energy from fires"?

Lines 198 – 214, this is an interesting discussion. However, what I miss here is a discussion on the potential influence of the spatiotemporal progression of the fire season.

For example, the authors clearly find highest median FRP in more arid environments (e.g. southern Africa or interior Australia), these regions also tend to burn later in the fire season. So in Figure 2 when focusing e.g. on Australia. The increase in "fire size : median FRP" ratio isn't that simply because we are first looking at a dominant signal from tropical northern Australia and then the signal becomes more and more dominated by interior Australia towards the end of the fire season? In that light I like the suggestion of reviewer #1 to take an approach that has a stronger focus on vegetation types, or areas that are otherwise more similar in terms of climate and vegetation compared to the GFED regions.

Line 238 "Fire danger index has been constantly increasing during the last 50 years", I believe conclusions of that paper were a little more nuanced.

Figure 2: why do y-axis on the right side have no caption? Also, it's probably good to mention that "The background histograms represent the number of fire patches" in the caption. Finally, what is the size and ranges of the FRP-bins? Are you excluding bins with less than x fire patches?

Table 1: "FI at maximum size (MW)", seems to be incorrect since you did not look at the FI for the largest fires. Something like "FI with largest associated fire patch sizes", or similar may be more appropriate.

---

## Author Comment (AC1) · 15 Oct 2018

Major concerns:

I) The first is that the study is based on the GFED regions but then wants to interpret differences between the regions to be driven by biomass availability and drought. If you want to understand whether differences are driven by biomass and drought then a much more straight forward way to analyse the data would be to group them according to biomass and drought, not according to the GFED regions that average over all Northern Hemisphere Africa, which contains the whole gradient from desert with low biomass and strong drought to tropical rainforest with no drought and high biomass. I am very confident that the results could be much clearer and support your conclu-

sions much better if the study design was rearranged to directly look at the effects of drought and biomass on these relationships, by grouping the data according to these two parameters.

Answer: We agree that relying only on GFED regions tends to mix together biomes with different biomass, fuel types, and with very different drought conditions. The problem with the use of drought datasets is that it is difficult to choose how to perform the separation between different levels of 'drought severity' : we could focus on the length of the drought season, or the severity of the Fire Danger Index, a combination of both, etc .... This choice would seem quite arbitrary, and would require a dedicated analysis. Instead, we propose to use MODIS Land Cover Data to separate each GFED regions in different biomes (Forested, (green) Savannas (light green), Grasslands/Shrublands (orange), see Figure attached to the answer). We clearly see that the relationship varies with the biomes : the results are especially striking in Australia, where we see that the FRP/FS relationship differs a lot depending on the considered biome. Finally, since we do not directly study the relationship with biomass and drought, we removed from the abstract and the discussion the sentences where we claimed that the fire intensity was driven by these quantities.

Separating our analysis depending on land cover is also important regarding the second major concern that you have raised (and that has also been raised by the other reviewers). We considered that FRP could be used as a proxy of the fire reaction intensity in the flaming front, but we did not discuss the limitations of such an approach. Particularly, we realized that the reliability of this hypothesis strongly depends on the land cover : for grassland, most of the energy is released in the flaming front, whereas for forested areas, radiation from smouldering fires also contribute to FRP. Therefore, the separation into land cover also appears very natural, and we can now discuss the hypothesis 'FRP is a proxy of fire intensity' and the reliability of the results depending on the considered land cover.

Note that what we suggest here is a 'double' separation (GFED and Land Cover). We

think that keeping the separation into GFED regions (and separating each of them into different Land Cover) is important for two reasons : - They define regions which are widely used within the fire community - Grouping all fires belonging to a given biome without separating in GFED regions would mix together regions with different fire practices/policy/management

II) The second is that the manuscript needs a discussion of reliability of the data, especially for the fire intensity. There are a number of limitations on the observability of fire radiative power. The point that there are still these clear spatial and temporal patterns in my opinion indicate that there is useful information in the dataset, however the problems associated with the dataset should be mentioned and discussed. For instance the energy observed is the energy released in one pixel, this energy might come from a very intense fire covering a small part or a low intensity fire covering large part of the grid cell. The observation of fire intensity strongly depends on the scan angle. Moreover fire intensity has a diurnal cycle and peak fire intensity might differ between the biomes. The satellite overpass might happen at the peak time in some grid cells but not in others. Vegetation structure influences what the satellite can observe, intensity of sub-canopy fires will certainly be underestimated. I am not an expert in remote sensing, but I think that such issues need to be mentioned to provide a balanced discussion of the results.

Answer: A discussion on the data reliability was clearly missing. We realized that we did not defined well-enough what we meant by "fire intensity", and how does this relate to FRP. Moreover, we did not discuss or reference the spatial and temporal sampling error that might impact the measurements of FRP. We plan to do the following changes in the manuscript :

- First, we would like to replace in the text 'Fire Intensity' by 'Fire Radiative Power'. What we observe is FRP, and then we interpret it as a proxy of fire reaction intensity. We would also like to change the title of the article into : 'Varying relationships between fire radiative power and fire size at global scale' if the editor agrees.

- Second, we will provide a dedicated section in the discussion (with references) to thoroughly discuss these issues. Note that the separation into land cover strongly helps to discuss the reliability of FRP as a proxy of fire reaction intensity, since this is expected to depend on Land Cover. Here comes a draft of the dedicated discussion:

"In the previous section, we hypothesised that FRP could be used as a proxy of fire reaction intensity. We now focus on the limitations of such an approach. First, the energy released by a wildfire can be decomposed in three parts : convection, conduction, and radiation. FRP only represents the radiative part of the energy emitted by a fire. Moreover, the fire reaction intensity used in Rothermel's equation does not share the same spatial extent as FRP : fire reaction intensity pertains to the flaming front of the fire, while FRP integrates all the radiative energy emitted over a 1 km2 window. This means that radiation emitted from smouldering can also contribute to FRP, not only the flaming front. The impact should differ for different vegetation types : smouldering fires are more frequent in forested areas, whereas in grasslands most of the detected radiative power will be released by the active fire front. Another issue appears from the integration of radiative energy over the 1 km2 window : it is impossible to know if the detected FRP arises only from a fire covering the full 1 km2 area or only from a smaller fraction of the FRP pixel. However, we can expect this effect to be mitigated by the fact that our analysis does not account for very small fires, since the FRY database does not provide fire patches smaller than 107 ha for MCD64A1. Finally, a recent study (Roberts et al. 2018) used 3D radiative transfer simulations to show that the canopy structure intercepts part of the FRP emitted by surface fires. This means that the FRP measured from remote sensing for forested areas and savannas could underestimate the real FRP. We can also expect this underestimation to vary with tree species. For example, it is probable that the amount of radiation energy intercepted by the canopy differs strongly between canopy fires from highly flammable black pines from BONA (Rogers et al. 2015) and surface fires from pine needle bed in BOAS. All these considerations emphasize the importance to split the study of the relationship between fire size and FRP in different vegetation types, since the reliability of using FRP as a proxy
of fire reaction intensity depends on it."

"The amount of radiative energy reaching the MODIS instruments is much smaller at large scan angles than at Nadir. This means that the MODIS instruments will be less sensitive to low values of FRP at high latitude (Giglio et al. 2003, Schröder et al. 2005). This could explain the difference of the distribution of FRP associated with fire patches in BONA (Figure 2) : the stronger asymmetry of the distribution in this region (i.e. the larger tail toward high FRP values) could arise from missing active fire data from less intense fires in this region. The temporal sampling of FRP also differs with the latitudinal coordinate : the number of satellite overpass is larger at high latitude than at the equator (from 2 observations per day until 15 at the poles, Giglio et al. 2006). This should rise the probability to recover FRP information for fire patches at high latitude, assuming that their radiative intensity is high enough to exceed the higher detection threshold at larger scan angles. Also, in some regions (such as NHAF and SHAF) fires exhibit a strong diurnal cycle (Giglio et al. 2006). The detection rate of active fires will therefore be higher if the peak of diurnal intensity is synchronized with satellite overpass. However, we can expect the sampling error rate and the variation of FRP sensitivity with latitude to be more homogeneous within each GFED regions that at global scale."

Please find our point-by-point answers to specific comments in the following.

1) l.17: thresholds differ between regions: what defines the regions? climate, humans, vegetation types?

Answer: We meant GFED regions. As stated in our main answer to the review, we also separate each GFED regions in different biomes (see Figure 3). We have added in the abstract that the relationship changes with the region and the considered vegetation type.

2) l. 20: seasonal effects, could there be an influence of anthropogenic fire use too? Percolation theory explains why fires are most intense or why fires are smaller in the

late season? I guess the latter.

Answer: Yes. For example, in the discussion (l. 252), we mention the use of prescribed burning at the beginning of the fire season in Africa to limit fire size. Concerning the percolation theory, the sentence in the abstract was not clear enough. Indeed, the term "this effect" was referring to the decrease of fire size toward the end of the season. We have modified the text to make it clearer.

3) l. 25-27: not sure I agree 100% with the reasoning: fire models have been included before in DGVMs, for instance Arora and Boer (2005). I think the reason was more the strong impact on vegetation and overestimation of tree cover in savannas in many DGVMs.

Answer: We agree. We removed 'As a result' from the sentence.

4) l. 28: prediction of vegetation dynamics and the carbon cycle.

Answer: The text has been changed.

5) l 47: also the impact of fire varies with the size, the fire size characteristics therefore could be more informative than only burned area.

Answer: Yes, we agree that the shape of the fire can have an effect on its impact on the vegetation. We have added a couple of references (Greene et al. 2005, Cary et al. 2009), and we have added a sentence in the text to mention this effect.

6) l. 52: maybe drivers of propagation and ignition are not driven by the same climate variables, but the fraction of ignitions turning into fires is determined by similar drivers, burned area and fire counts therefore have quite similar spatial patterns.

Answer: We agree, but in this paragraph, we do not discuss/compare Burned Area to fire counts. We just stated that separating ignitions from propagation would bring more information than just using BA or fire counts. However, this comment is related to comment 14 (where we actually compared fire counts and fire ignition), which brought

some modification in the manuscript.

7) l.57: is it fire intensity of fire line intensity? and based on the equations this is not expected for large fire size? or do you mean the Rothermel equations were only tested for small scale (laboratory to stand scale) and it is unclear whether the equations hold true for larger scales (not for larger fires).

Answer: Actually, this is reaction intensity of the fire front. We meant that Rothermel's equation was only tested at local scale, as stated at l. 38-39. We have modified the text to recall this on l. 57.

8) l. 95: explain the difference between fire intensity and fire radiative power. Answer: We will provide a more accurate definition of fire (reaction) intensity and fire radiative power when the terms are introduced in the manuscripts (i.e., at line 57 for fire intensity and line 85 for FRP). Following the received comments, we have decided to focus on FRP throughout the presentation of the results, and introduced the use of FRP as proxy of FI in the interpretation of the result only. The limitation of this hypothesis is discussed in the discussion section.

We added also some references on the use of FRP as a proxy for several fire severity applications to justify the use of this index in our analysis:

-field work to estimate fire intensity on fire severity and impact on soil: Barret & kasischke 2013, Sparks et al. 2018 (in biogeosciences).

-fire risk modelling of fire size and intensity based on FRP: Hernandez et al. (2015)

-biomass combustion rates from FRP in Africa: Roberts et al. (2005)

-relating a fire spread equation (Byram) to fire intensity from infrared remote sensing: Johnson et al. (2017)

9) l. 96: are there any spatial or temporal patterns in the the discarded fire patches ? This might indicate biases in the FRP detection.

Answer: Yes, there are some spatial patterns. We now discuss this in the methodology and discussion section, and we have added a couple of supplementary plots with :

- a map of the ratio of missed matches between fire patches and active fire pixel data.

- a histogram showing the global fire size distribution of fire patches and the distribution of fire patches without recovered active fire information.

10) l.110: text says median, figure caption says mean. The figure could also include a burned area map to show that the patterns are different, between the characteristics.

Answer: This was a typo. We have added a map of yearly burned area over the same type period as MCD64A1, and briefly discuss the difference with fire count in the text.

11) l.112: patterns of size and FRP look not so similar to me.

Answer: We have modified the text, and try to provide a more accurate description of the figure.

12) l. 117: use either mean or average.

Answer: This was a typo. We kept 'average'.

13) l.119: could this peak simply be because lower intensity is simply not detected by the satellite. What is the explanation for this peak at intermediate fire intensities?

Answer: Thresholds of FRP detection vary between 9 and 11 MW (Schroeder et al. 2010, Roberts and Wooster 2008) for MOD14 and MYD14, below which no data are available. In turn, remotely sensed finer resolution analysis actually concluded that MOD14 may underestimate by 20% captured fire pixels, particularly for small fires (Wooster et al. 2012, Peterson et al. 2013). Beside spatial resolution, different sensors can differentially capture FRP (Li et al. 2018) due to solar angle and vegetation types. The 9-11MW threshold falls in the 1st bin of FRP in Figure 2, and could therefore explain the peak at intermediate FRP.

We are now discussing this in the manuscript (in the result and discussion section), and we have added the aforementioned references.

14) l.121: change "number individual.." to " number of individual..." I assume fire counts is related more closely to burned area as two counts could be individual fires or the same fire, so some differences are also expected.

Answer: We agree that active fire counts are closer to burn area than individual number of fire patches. We removed the sentence from the text.

15) l.124: It would be useful to consistently use FI or FRP, now it is FI in the text and FRP in the figure.

Answer: Following the comments of the reviewers, we choose to use FRP throughout the text. We agree that we interpret FRP as a proxy of FI, not directly as FI itself. Moreover, we added a paragraph in the discussion about the differences between reaction intensity and FRP (see our main answer).

16) l. 125-28: I don't see that the fire size is clearly decreasing. it is a bit tempting also to interpret the error bars as error bars. Maybe having three lines for 25th percentile, median 75th percentile could avoid that misunderstanding. Probably showing the 4th order polynomial with uncertainty bands could give a better impression whether decreases and maximum are robust. My confidence based on the plots shown is rather low, and now these threshold become quite important for the following discussion. Showing some kind of robustness and uncertainty on this threshold would therefore be important.

Answer: The problem with lines is that it makes it difficult to represent the burn date information encoded with the dot/colorbar color, which is an important information for the discussion. We prefer to keep the plot as it is now. However we have added the interpolated polynomial "under" the dot, to show the humped relationship of the median fire size wrt FRP. Also, please note that the large range of fire size (due to the

75th quantiles, 5 to 10 times bigger to median fire size) render difficult the sight of the decrease after the threshold is reached.

We also realized that the description of our methodology was not accurate enough. First, we only fit the median value of FRP, not the total distribution. The polynomial fit is only used to smooth the maximum median FRP value from the FRP vs FS relationship. We then perform two linear regression : one in the range [FRP > 0, FRP < FRPmax], one in the range [FRP > FRPmax, FRP < 300]. We obtain uncertainties (with their correlation) on each of the 5 parameters of the polynomial fit, and the uncertainty on the each slope of the linear regression. The uncertainty on the parameters of the polynomial are low (less than 1% in terms of relative uncertainties for all parameters). Similarly, the relative error on the slope fitted over the range [FRP > FRPmax, FRP < 300] is lower than 1%.

17) l. 174: higher FI threshold for forests: can this be explained by Rothermel?

Answer: Rothermel only explains the expected linear relationship between fire size and fire intensity. A higher FRP threshold simply means that Rothermel s equation is valid on a larger range of FRP, and we suggest that this could arise from the fuel array continuity.

18) l. 176-7: Rothermel also uses different parameters for different vegetation types and fuel moisture and is able to reproduce the varying constraint hypothesis.

Answer: Yes, Rothermel is able to simulate the varying constraint hypothesis. but we show that Rothermel is no more valid (or highly affected by another factor; potentially landscape fuel continuity) for high intensity fires. For example, late fire season in Africa is dry and not fuel-limited so this season should experience the highest FS as a function of FI. We believe this paragraph is not fully related to our results, so, for clarity of the manuscript and regarding the comment of the reviewer, we removed the paragraph concerning this comment.

[Figure]

19) l.185: GFED regions are not biomes

Answer: This is true. We have now separated each GFED regions in different biomes using Land Cover data from MODIS. See our main answer to the review.

20) l.195: so you expect a lower threshold for higher fragmentation? is that what you find in your analysis?

Answer: We fully rephrased and developed this sentence. We mention here that landscapes can be intrinsically fragmented (by roads, or cropland mosaic), and seasonally fragmented by successive fires. In turn, successive fires can interact at the landscape scale, so the edge of the first fire acts as a barrier for the second fire propagation (Teske et al. 2012). In savannas, patchy mosaics of burned land are then intentionnaly created early in the fire season as a preventative strategy for large fires emerging later in the season (Laris et al. 2002). This sentence is part of the discussion and we propose here an hypothesis on our findings of the FS/FRP threshold.

21) l.215: I would expect a very high fragmentation in EURO and TENA (lots of big streets) and strongly managed, which is why fire models usually overestimate burned area there. Is this only meant for interpreting the seasonality? so no fragmentation due to burning?

Answer: In these regions, despite being highly populated and urbanized, burned area has been shown to be mostly driven by weather and anthropogenic process rather than landscape fragmentation (cf figure 9 in Le Page et al. (2015)), a result supported by our findings. We added this reference in the manuscript.

22) l.220: not all the tropics has rainfall all year long. Answer: We agree. We have changed the text to 'where drought period are shorter'

23) l. 221: are you suggesting that the savanna species suppress fire? burned are is much higher in savannas than in TENA and BOAS. I don't understand the logic here.

Answer: We are speaking about propagation once a fire has started. Our results show

that fires are larger in forests of TENA/BONA than in savannas. However the number of individual fire patches is much lower in TENA/BONA, which results in a lower burn area.

In BONA/TENA, once a fire has started (mostly on the ground layer), they turn into crown fires which are hardly controlled, while grassland fires can be more easily stopped by changes in weather or landscape obstacles. We rephrased the sentence in the manuscript:

'They can therefore propagate further than ground fire and fire resistant species found in savannas and woodlands in semi-arid tropical regions' => 'they can therefore propagate further than herbaceous fires hardly turning into crown fires in savannas and woodlands in semi arid tropical regions.'

24) l. 223: vegetation is less flammable where?

Answer: In BOAS. We have modified the text.

25) l. 237: FDI increased everywhere?

Answer: Rather than FDI, the article actually focus on fire season length. This has been obsereved in some regions only. We have changed the text.

26) l.251: agricultural expansion leads to a reduction of burnable area. why? Croplands are burned, pastures are burned. Also the more fragmented landscape, is there a study showing that the landscape is more fragmented. I is an assumption in models and to explain the decline in burned area. Give a reference where this fragmentation is observed, or identify it as a common assumption. Could this be an effect of having smaller fires in croplands and therefore the detectable burned area is declining, not the burned area itself?

Answer: We now provide in the manuscript some references illustrating the fragmentation of landscapes in savannas worldwide :

- Sulieman, HM. 2018. Exploring drivers of forest degradation and fragmentation in sudan: the case of Erawashda forest and its surrounding community. Science of the total environment. 621: 895-904.

- Oliveira SN, de Carvalho OA, Gomes RAT, Guimaraes RF, McManus CM. 2017. Landscape-fragmentation change due to recent agricultural expansion in the Brazilian Savanna, Western Bahia, Brazil. regional environmental change 17(2): 411-423

- Kamusoko C., Aniya M. 2007. Land use/cover change and landscape fragmentation analysis in the Bindura District, Zimbabwe . Land degradation and development 18(2): 221-223

27) l. 259: BA saturates toward the end of the drought season: is this really reproduced by models? Any reference?

Answer: We did not find any references about this, but this mechanism looks realistic in regions with high BA (NHAF/SHAF) looking at the equations from Thonicke et al. We rephrase the sentence : "Because of the reduction of the available fuel load due to burning by preceding fires, we can expect than BA saturates toward the end of the drought season in DGVMs."

28) l. 267: fire-prone ecosystems: actually you didn't group the analysis by ecosystmes, for the tropical regions you group everything together, tropical rainforest and savannas are not separated. I think it would be smarter to group the data for this analysis based on vegetation and climatic parameters not by geographical regions. grouping high and low tree density together could confound the results.

Answer: We agree. See our main answer to the reviewer's comment.

29) l. 271: FI threshold is driven by biomass and drought severity: Most of the regions have a strong variation of biomass and drought severity. It therefore would be better to use drought severity and biomass to group the data.

Answer: See main answer to the reviewer's comment. We suggest to divide each

region depending on their land cover.

[Figure]

[Figure]

**Fig. 1.** FRP vs fire size for different biomes

[Figure]

**Fig. 2.** Map of the land cover biomes

---

## Author Comment (AC2) · 15 Oct 2018

Major concerns:

This manuscript is interesting and fit well with the focuses of BG. It can be published after a careful revision. I am not a fire ecologist. Consequently, I met a lot of difficulties in understanding concepts, variables names and their definitions you used in the manuscript. The guiding principle of your analysis is the Rothermel (1972)'s fire spread model ("Rothermel's equation" line 35, "Following the hypothesis from Rothermel's equation of fire spread" line 170). It is a very detailed local scale model. It is one of the most used models to simulate the forward rate of spread at the front of a surface fire, and is the primary fire spread model applied in many fire prediction systems. In the

Rothermel's model, rate of spread is simulated as a function of topography, microclimate conditions and a fire behavior fuel model or fuel model that consists of numerous parameters for a given fuel complex. Standard fuel models have often been shown inappropriate for representing local conditions. In this manuscript, you referred to the Rothermel (1972)'s equation. In the original USDA paper, the number of equations was c.a. 90. It will be fine that in your up-scaling procedure, from local to global, you explain how you summarized the Rothermel (1972)'s fire spread model for finally analyze the relationship between fire patch area and fire intensity. A short explanation will be useful and will clarify the discussion in which you mixed: fuel biomass availability, biomass gradient, moisture content of the fuel, fragmentation, wind speed, fuel bulk density, fuel load, etc.

Answer: Looking at your review and the reviews from the two other referees, we realized that we did not defined well the different quantities that we are using throughout the article : fire (reaction) intensity, FRP, and how they relate to Rothermel's equation. We are now giving a more careful definition of all terms. Concerning Rothermel's equation, we are using the "main" equation for Rate of Spead, which is used in most fire module : this is equation (10) in the Rothermel (1972) article. We will clarify this in our revision, because our description is not straightforward for anybody without a fire ecologist background. Also, note that following the suggestions from all referees, we decided to change "Fire Intensity" with "Fire Radiative Power" in the title and throughout the article. We are looking at relationships between fire size and FRP, and then we interpret FRP as a proxy of FI.

My second main concern is your cutting of continents by using the one proposed by the GFED. The 14 regions are very arbitrary. As an example EURO includes the surrounding of the northern part of the Mediterranean Sea where the fire regime surely doesn't follow the same pattern than in more Northern regions. Likely using a more "ecologically-based" or "climatically-related" cutting will yield contrasted results?

Answer: Yes, this was a request from the three reviewers.

[Figure]

Splitting the data following drought is difficult, because there are plenty numerous possibilities to perform the split (using the length of the season ? The intensity of the drought index ? A combination of both ?). We suggest to split each GFED regions using MODIS Land Cover information: this will allow not to mix together grassland,savannas and forests in Africa for example. The results are really striking in Australia, where the relationships strongly differs between different land cover types.

Also, we realized that the reliability of the hypothesis which claims that FRP could be used as a proxy of fire intensity depends on the land cover: FRP integrates all the radiative energy from the fire, from the flame front or from smouldering. Only the flame front is related to the rate of spread. In grassland, the flame-front will be the main contribution to FRP, but this is not always the case for other land cover. Therefore, separating our analysis depending on land cover also allows us to discriminate areas where FRP is a reliable proxy of fire intensity.

Please find our point-by-point answers to specific comments in the following.

1) Line 23 plant biomass distribution.

Answer: The text has been changed.

2) Line 25 rather ecological driver than climatic variable.

Answer: Through its effects on the carbon cycle, fire is an important climatic process. But this is true that fire is important for both ecological and climatic effects (as described later in the introduction). Since in this first paragraph we focus on climate modeling, we prefer to keep the sentence as it is.

3) Line 29 reliable burned area, active fires and fire intensity global dataset.

Answer: The text has been changed.

4) Line 45 fire patches vs raw burn area. Please could you explain?

Answer: Burned area integrates all fire patch areas into a single value. Recent studies

now split analysis of the total burned area into patch level analysis allowing for a more precise information on ignitions and fire spread processes underlying the final burned area.

5) Line 54 please define BA here (burned area).

Answer: Actually we first used the term Burned Area on l. 30. We now introduce the term BA there.

6) Line 62 please detail MCD14ML. Best to give the complete name of the remotely sensed products you used and their DOI if available.

Answer: I will detail the dataset, and try to find the DOI.

7) Line 76 fire patch size why not fire patch area ?

Answer: We tried to keep the same terminology as in Laurent et al. 2018.

8) Line 74 "validated against Landsat fire polygons".

Answer: The text has been changed.

9) Line 77 Standard Deviation Ellipse (SDE) Please could you explain how this parameter calculated? It does not seem further used in the manuscript except lines 87 and 89. One SDE covers approximately 68 percents of the fire patch. You applied a cutoff at SDE + 1 km, why not 2 SDE?

Answer: The SDE were obtained with the "aspace" R package. We have modified the text (l. 80). Taking 2 SDE as the matching radius with FRP pixel would have yield lots of double association in the database. We prefer to be conservative, even if this result in a reduction of the usable number of fire patches.

10) Line 90 30-day buffer seems very long. During this delay surface reflectance may drastically change with resprouter shrubs or some bunchgrasses.

Answer: We agree, but this corresponds to the high uncertainty on the burn scar detection from BA dataset. Moreover, we did some test by reducing the 30 day buffer, and this had no significant effect on the result (it only reduces the number of patch with associated FRP).

11) Line 95 you wrote "In this analysis, we used FRP as a proxy of fire in tensity, later called FI". Further we still found FRP in the text and in the graphs.

Answer: Following the remarks of the other referees, we prefered to change FI into FRP throughout the text.

12) Line 112 "Brazilian tropical savannas". On fig 1b, most red dots are located across Argentina and not across Brazilian tropical savannas!

Answer: Yes, we agree. We have changed "Brazilian tropical savannas" to "Patagonia".

13) Line 125 please define the meaning of GFED. Please use the full names of the regions in Table 1.

Answer: We have inserted the full name on line 120, the first time GFED is mentioned.

14) Line 126 fitted rather than interpolated.

Answer: We have changed the text.

15) Line 130 humped relationships in CEAM, EQUAS, SEAS. This type of "humped" relationships seems to occur elsewhere? You presented these three areas as equatorial biomes. This means closed to equator or with a particular climate pattern? (See my previous comment on your geographical cutting).

Answer: We now separate each GFED region depending on their vegetation types, using land cover from MODIS. This has yield a new paragraph in the Methodology and the Results section.

16) Line 139 MW-1

Answer: Text has been changed.

17) Line 206 percolation or cellular automata?

Answer: We meant percolation.

18) Figure 2 FI in the figure legend and FRP in the x-axis. Y-axis scales drastically change depending of geographic area and so complicate the reading.

Answer: This was a mistake, we have modified the text.

19) Figure 3 are you sure that this figure is necessary (see Table 1 content).

Answer: We agree, especially now that we are separating GFED regions in multiple biomes. We have removed the figure.

[Figure]

**Fig. 1.** FRP vs fire size for different biomes

[Figure]

**Fig. 2.** Map of the land cover biomes

---

## Author Comment (AC3) · 15 Oct 2018

Major concerns:

I) The manuscript lacks a proper discussion (and references) of potential issues that may arise when estimating fuel consumption and subsequently fire intensity from FRP. FRP observations from MODIS represent infrequent snapshots of energy release across the pixel area (at best âĹ́ij1km2 at nadir). This results in a number of difficulties when linking FRP to fire temperatures or -intensity- of which several will likely be a function of environmental gradients. First, FRP is an estimate of energy release across an entire pixel, âĹ́ij1 km2 at nadir for MODIS. It is very uncertain what fraction of the grid cell is actually burning (and this is likely a function of fuel loads and other aspect

of fire behavior). Yet, this is a requirement to estimate fire intensity because if 1% of the pixel produces 10 MW of energy, or 50% of the pixel produces the same amount makes a difference of 50 times the "intensity". Second, several studies suggest that vegetation structure (in particular tree cover) also have a significant effect on the relationship between fuel consumption and observed FRP (e.g. Roberts et al., 2018 RSE). Third, the sensitivity of the MODIS instruments to detect active fires (i.e. minimum FRP that can be observed) is a direct function of the scan angle and is up to a factor of 5 lower at large scan angles compared to nadir. This may be important when looking at distributions (e.g. median), because you are likely to strongly underestimate the occurrence of low FRP values. Fourth, the fire diurnal cycle (a function of fuel conditions, vegetation type, and climate) also produces a sampling error, since there are only few daily overpasses and in some ecosystems fire activity may peak already early in the morning while in others this maybe later in the afternoon. It would be important to properly discuss what "MODIS FRP" actually represents. I also disagree with the statement "This is in agreement with .. , since these quantities are two proxies of the number of ignitions." (lines 120-122). I do not see how the number of active fire detections is related to ignitions? A single fire may produce up to hundreds of active fire (FRP) detections if it becomes large enough and burns for a long period of time. Several studies have linked active fire detections (with or without FRP) to total amounts of fuel consumption (or biomass burned), which would be a function of area burned, fuel loads and other conditions. Moreover, looking at the distribution of FRP detections may become problematic here. In high fuel load temperate and boreal forested systems a large share of the active fire detections may come from smouldering rather than the active fire front (and ratios may change over the fire's lifetime), while for grasslands it may be mostly actively flaming fire fronts that are observed. In this light it would be important to much better define "fire intensity" (i.e. what do the authors want to measure exactly?), and discuss how using FRP as a proxy for this quantity may be further influenced by the above mentioned limitations.

Answer: We highly agree with the numerous comments on the use of FRP stated by

the reviewer. First, using FRP as a proxy of Fire reaction intensity is not straightforward, and we agree that it should depend on the considered land cover. We decided to replace all occurrences of 'Fire Intensity' in the text with 'Fire Radiative Power' (including the title), since this is what we are really observing in the analysis. Then, we discuss our result under the hypothesis that FRP could be used as a proxy of fire reaction intensity (that we now clearly define in the text), and we discuss the limitation of such an approach. Note that we are now dividing each GFED regions into different land covers (forests/savannas/grasslands, see minor comment 9)). This separation is really helpful, because we can now separate in each GFEd regions grasslands (where FRP can be safely used as a proxy of fire intensity) from forests (where canopy could intercept part of the emitted radiation, and where smouldering could significantly contribute to the detected FRP). We added references to support the discussion.

We have also added in the discussion a paragraph (and references) about the spatial and temporal errors of MODIS instruments. We also think that keeping the separation into GFED regions allows to mitigate the sampling error : for example, if detection threshold varies with latitude, we can expect it to significantly differ between BONA and NHAF, but to vary less within each region. We then expect FRPs to be equally related to FI within a biome, whatever the FRP intensity, so the relationships with FS is conserved in our results. This uncertainty in FRP was also pointed out by reviewer 1 for which we also provide additional information and references on detection thresholds. In the following, we displays a draft of the paragraphs we would like to add to the discussion section:

"In the previous section, we hypothesised that FRP could be used as a proxy of fire reaction intensity. We now focus on the limitations of such an approach. First, the energy released by a wildfire can be decomposed in three parts : convection, conduction, and radiation. FRP only represents the radiative part of the energy emitted by a fire. Moreover, the fire reaction intensity used in Rothermel's equation does not share the same spatial extent as FRP : fire reaction intensity pertains to the flaming front of the

fire, while FRP integrates all the radiative energy emitted over a 1 km2 window. This means that radiation emitted from smouldering can also contribute to FRP, not only the flaming front. The impact should differ for different vegetation types : smouldering fires are more frequent in forested areas, whereas in grasslands most of the detected radiative power will be released by the active fire front. Another issue appears from the integration of radiative energy over the 1 km2 window : it is impossible to know if the detected FRP arises only from a fire covering the full 1 km2 area or only from a smaller fraction of the FRP pixel. However, we can expect this effect to be mitigated by the fact that our analysis does not account for very small fires, since the FRY database does not provide fire patches smaller than 107 ha for MCD64A1. Finally, a recent study (Roberts et al. 2018) used 3D radiative transfer simulations to show that the canopy structure intercepts part of the FRP emitted by surface fires. This means that the FRP measured from remote sensing for forested areas and savannas could underestimate the real FRP. We can also expect this underestimation to vary with tree species. For example, it is probable that the amount of radiation energy intercepted by the canopy differs strongly between canopy fires from highly flammable black pines from BONA (Rogers et al. 2015) and surface fires from pine needle bed in BOAS. All these considerations emphasize the importance to split the study of the relationship between fire size and FRP in different vegetation types, since the reliability of using FRP as a proxy of fire reaction intensity depends on it."

"The amount of radiative energy reaching the MODIS instruments is much smaller at large scan angles than at Nadir. This means that the MODIS instruments will be less sensitive to low values of FRP at high latitude (Giglio et al. 2003, Schröder et al. 2005). This could explain the difference of the distribution of FRP associated with fire patches in BONA (Figure 2) : the stronger asymmetry of the distribution in this region (i.e. the larger tail toward high FRP values) could arise from missing active fire data from less intense fires in this region. The temporal sampling of FRP also differs with the latitudinal coordinate : the number of satellite overpass is larger at high latitude than at the equator (from 2 observations per day until 15 at the poles, Giglio et al.

2006). This should rise the probability to recover FRP information for fire patches at high latitude, assuming that their radiative intensity is high enough to exceed the higher detection threshold at larger scan angles. Also, in some regions (such as NHAF and SHAF) fires exhibit a strong diurnal cycle (Giglio et al. 2006). The detection rate of active fires will therefore be higher if the peak of diurnal intensity is synchronized with satellite overpass. However, we can expect the sampling error rate and the variation of FRP sensitivity with latitude to be more homogeneous within each GFED regions that at global scale."

Please find our point-by-point answers to specific comments in the following.

1) Line 87. That is ok, but what do you do if you have two adjacent fire patches? Are you double counting the active fire detections?

Answer: Yes, this is what we do. However, note that we are performing the matching using Standard Deviation Ellipses (SDEs) from the fire patches, since these are the information provided by the FRY database. SDEs delimit 2/3 of the burn pixel of the fire patches, and are localized around the central area of the patch. This should limit the amount of attributing twice a active fire pixel.

2) Line 95. ".., we compute for each patch the mean FRP value of all .. ". This isn't entirely clear to me, do you first estimate the mean of each patch and then look at the median across patches? Again, it would be important to understand what the distributions look like (e.g. across land cover types) to understand the potential implications of such decisions.

Answer: Yes, this is what we do. We have added as supplementary plots :

- a map of the ratio of missed matches between fire patches and active fire pixel data.

- a histogram showing the global fire size distribution of fire patches and the distribution of fire patches without recovered active fire information.

We now discussed these plots in the discussion, they help us to discuss the limitation

of using FRP as a proxy of Fire Intensity.

3) Line 155 "In each 1x1 cells", typo.

Answer: The mistake has been corrected.

4) Lines 155 – 160, please move this to the methods section, accompanied by a short explanation on how that helps to answer your research questions.

Answer: We have moved the section to the methodology section.

5) Line 170 "Following the hypothesis from Rothermel's equation", maybe be a bit more specific here and add references. For clarity you could also repeat your own objectives here, e.g. "We aim to investigate if fire size and intensity are driven by a same set of environmental and climate conditions.." Also, I am somewhat surprised that in addition to speed, the authors don't mention fire duration as a potential driver of larger fire sizes.

Answer: We agree. This is also related to the first concern of reviewer 2. We are now giving more details about Rothermel's equation.

6) Line 174 "Tropical areas" is not a vegetation type, delete?

Answer: We meant tropical forest. We now separate each GFED regions in biomes.

7) Line 178 "experience limited fire energy" what does this mean? Do you mean to say something like "In equatorial areas with high annual rainfall, biomass burning is characterized by low spread rates are combustion completeness (cite), resulting in a more gradual release of energy from fires"?

Answer: Yes, this is what we meant. Also, if the fuel is not totally dry, part of energy release by the fire will be 'wasted' to vaporize the remaining water in the fuel.

9) Lines 198 – 214, this is an interesting discussion. However, what I miss here is a discussion on the potential influence of the spatiotemporal progression of the fire season. For example, the authors clearly find highest median FRP in more arid environments

(e.g. southern Africa or interior Australia), these regions also tend to burn later in the fire season. So in Figure 2 when focusing e.g. on Australia. The increase in "fire size : median FRP" ratio isn't that simply because we are first looking at a dominant signal from tropical northern Australia and then the signal becomes more and more dominated by interior Australia towards the end of the fire season? In that light I like the suggestion of reviewer #1 to take an approach that has a stronger focus on vegetation types, or areas that are otherwise more similar in terms of climate and vegetation compared to the GFED regions.

Answer: We agree. We are now separating each GFED regions in biomes. Note also that the separation into biomes is extremely helpful when it comes to the discussion related to your major concern for our analysis.

We put here the main answer to reviewer #1 about separating GFED regions in different land cover.

"We agree that relying only on GFED regions tends to mix together biomes with different biomass, fuel types, and with very different drought conditions. The problem with the use of drought datasets is that it is difficult to choose how to perform the separation between different levels of 'drought severity' : we could focus on the length of the drought season, or the severity of the Fire Danger Index, a combination of both, etc .... This choice would seem quite arbitrary, and would require a dedicated analysis. Instead, we propose to use MODIS Land Cover Data to separate each GFED regions in different biomes (Forested, Savannas, Grasslands/Shrublands, see Figure attached to the answer). We clearly see that the relationship varies with the biomes : the results are especially striking in Australia, where we see that the FRP/FS relationship differs a lot depending on the considered biome. Finally, since we do not directly study the relationship with biomass and drought, we removed from the abstract and the discussion the sentences where we claimed that the fire intensity is driven by these quantities."

10) Line 238 "Fire danger index has been constantly increasing during the last 50

years", I believe conclusions of that paper were a little more nuanced.

Answer: Actually, the article focused on the length of the fire season rather FDI. The author claims that the global fire season length has increase, even though fire season length can still decrease in some areas of the world. We have modified the text.

11) Figure 2: why do y-axis on the right side have no caption? Also, it's probably good to mention that "The background histograms represent the number of fire patches" in the caption. Finally, what is the size and ranges of the FRP-bins? Are you excluding bins with less than x fire patches?

Answer: We have added a caption on the y-axis, which represent the number of fire patches in each FRP bin (corresponding to the background histograms). We also modified the caption and give the ranges of FRP bins in the caption and in the text.

12) Table 1: "FI at maximum size (MW)", seems to be incorrect since you did not look at the FI for the largest fires. Something like "FI with largest associated fire patch sizes", or similar may be more appropriate.

Answer: Yes, we agree. We have modified the legend and the table. We have also found some similar occurrences in the text, and we have changed them.

[Figure]

**Fig. 1.** FRP vs fire size for different biomes

[Figure]

**Fig. 2.** Map of the land cover biomes

---

## Author Comment (AC4) · 16 Oct 2018

Please find our point-by-point answers to specific comments in the following.

1) L35. Why is Van Wagner cited in relation to Rothermel's model, with which he had no relation whatsoever? Van Wagner was Canadian, and so involved with the Canadian fire behaviour prediction system, not the U.S.

Answer: This was a mistake. We removed the reference to Van Wagner.

2) L36-37. "whose rate of spread scales with a power function of the wind velocity, landscape slope and fire intensity." The authors are referring to reaction intensity, not fire intensity (aka fireline intensity, which is the product of rate of spread, fuel consumption and heat of combustion and can be correlated to a certain extent with FRP).

Answer: Yes. This was also a remark from all reviewers. We did not define clearly what we meant by fire intensity. We now explicitly say in the text that this is fire reaction intensity of the flaming front.

3) L41-42. "On the other hand, the velocity of fire propagation determines the amount of fuel entering the combustion zone, and therefore feeds back on the intensity of the fire event." Not sure what this means. Rate of spread is an intrinsic component of fire intensity but not because it affects fuel consumption.

Answer: We meant that a fire need 'new' fuel to continue to burn. There is therefore a feedback between fire intensity and rate of spread: an intense fire is more likely to propagate faster, therefore to have more fresh fuel entering the combustion zone, therefore to continue burning, etc ...

4) L42-43. "fire intensity also significantly impacts the fuel combustion completeness". It's the other way around, fuel consumption is an element in the calculation of fire intensity.

Answer: You are right. This was a mistake, and we removed the sentence from the text.

5) L57. This is general, i.e. not specific of Rothermel 0 s model. For given fuel conditions/ fuel types faster fires are more intense, and faster fires will become large.

Answer: Yes this is true. We have changed the text.

6) L95. Has fire intensity been defined?

Answer: See answer to comment 2.

7) L170. The hypothesis does not stem from Rothermel's model, it just happens that fire intensity by definition (Byram 1959) is the product of rate of spread, fuel consumption and heat of combustion, as mentioned before.

Answer: Yes, this is true that this effect rather depends from Byram definition of fire

intensity, not from Rothermel's model. We will modify the text (and also mention that we used the Byram definition of fire intensity).

8) L221. "They can therefore propagate further than ground fire and fire resistant species found in savannas and woodlands". This sentence is confusing. Fire in savanna is driven by grass, not by trees (which are resistant only in the sense that they are fire adapted).

Answer: We realized that this sentence was not clear. We rephrased it in the manuscript: 'They can therefore propagate further than ground fire and fire resistant species found in savannas and woodlands in semi-arid tropical regions' => 'they can therefore propagate further than herbaceous fires hardly turning into crown fires in savannas and woodlands in semi arid tropical regions.'

9) I think the interpretation of the findings, by being concentrated on the effect of fuel connectivity, is restrictive. The authors could improve the discussion by considering that the most powerful driver of fire spread/size is wind speed (see the switches of Bradstock 2010). Thus, fuels can be totally available to burn due to drought, and produce intense fires that are not that large because they do not coincide with strong winds and low relative atmospheric humidities. Thus, the annual cycle of fire extent and intensity is also a matter of timing of coincidence between drought and atmospheric conditions.

Answer: We agree, but this would require a dedicated analysis using wind power/wind orientation datasets. We will mention this in the discussion.